# EXPLANATORY LEARNING: BEYOND EMPIRICISM IN NEURAL NETWORKS

## ABSTRACT

We introduce Explanatory Learning (EL), an explanation-driven machine learning framework to use existing knowledge buried in symbolic sequences expressed in an unknown language. In EL, the burden of interpreting explanations is not left to humans or human-coded compilers, as done in Program Synthesis. Rather, EL calls for a learned interpreter, built upon existing explanations paired with observations of several phenomena. This interpreter can then be used to make predictions on novel phenomena, and even find an explanation for them. We formulate the EL problem as a simple binary classification task, so that common end-to-end approaches aligned with the dominant empiricist view of machine learning could, in principle, solve it. To these models, we oppose Critical Rationalist Networks (CRNs), which instead embrace a rationalist view on the acquisition of knowledge. CRNs express several desired properties by construction, they are truly explainable, can adjust their processing at test-time for harder inferences, and can offer strong confidence guarantees on their predictions. As a final contribution, we introduce Odeen, a basic EL environment that simulates a small flatland-style universe full of phenomena to explain. Using Odeen as a testbed, we show how CRNs outperform empiricist end-to-end approaches of similar size and architecture (Transformers) for discovering new explanations of unexplained phenomena.

## 1 INTRODUCTION

Making accurate predictions about the future is a key ability to survive and thrive in a habitat. Living beings have evolved many systems to this end, such as memory (McConnell, 1962), and several can predict the course of complex phenomena (Taylor et al., 2012). However, no animal comes even close to the prediction ability of humans, which stems from a unique-in-nature system.

At the core of this system lies an object called *explanation*, formed by the proposition of a language, which has a remarkable property: it can be installed with ease into another human speaking the same language, allowing to make predictions on new phenomena without ever having experienced them. When the installation is successful, we say that the human has *understood* the explanation. This process is key to the success of human beings. An individual is not required to go through a painful discovery process for every phenomenon of interest, but only needs an operating system – mastering a language – and someone who communicates explanations; then, the individual can focus on unexplained phenomena. When an explanation is found for them, it is added to the existing shared collection, which we call *knowledge*.

How can we make machines take part in this orchestra? With this work, we try to shed new light on this problem. Specifically, we propose a learning paradigm to allow machines (i) to *understand* existing explanations, in the sense described above, and (ii) create new explanations for unexplained phenomena, much like human scientists do.

Our contribution in this sense is threefold:

i) We formulate the challenge of creating a machine that masters a language as a learning problem by treating explanations as simple strings, while keeping their role well separated from the role of the data. This results in the *Explanatory Learning* (EL) framework described in Sec. 2.

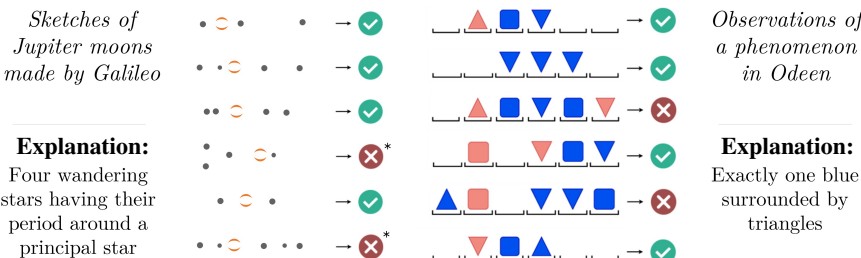

Figure 1: **The Odeen universe**. A convenient environment to study and test the process of knowledge discovery in machines. Like the night sky was for humans.[1]          *Galileo did not sketch negative examples.

ii) We present Odeen, a basic environment to test EL approaches, which draws inspiration from the board game Zendo (Heath, 2001). Odeen simulates the work of a scientist in a small universe of simple geometric figures, see Figure 1. We present it in Sec. 3, and will release it with this work.

iii) We argue that the dominating empiricist ML approaches are not suitable for EL problems. We propose *Critical Rationalist Networks* (CRNs), a family of models designed according to the epistemological philosophy pushed forward by Popper (1935). Although a CRN is implemented using two neural networks, its working hypothesis does not coincide with the model parameters. Instead, the working hypothesis of CRNs is a language proposition that can only be accepted or refused *in toto*. We will present CRNs in Sec. 4, and test their performance on Odeen in Sec. 5.

## 2    EXPLANATORY LEARNING

Humans do not master a language from birth. A baby can not use the message "this soap stings" to predict the burning sensation caused by contact with the substance. Instead, the baby gradually *learns* to interpret such messages and make predictions for an entire universe of phenomena (Schulz et al., 2007). We refer to this state of affairs as *mastering a language*, and we aim to replicate it in a machine as the result of an analogous learning process. In particular, we propose to do so without explicitly prescribing a grammar, using instead an incomplete set of example explanations in the form of plain sequences of symbols.

Our focus is on learning an interpreter purely from observations. This allows not only to make predictions about phenomena for which we already have an explanation, but also to *discover* an explanation for unexplained phenomena. We first describe the problem setup in the sequel, comparing it to existing ML problems; we detail our approach in Sec. 4.

**Problem setup.**    Formally, let phenomena $P_1, P_2, P_3, \ldots$ be subsets of a universe $U$, which is a large set with no special structure (i.e., the entire data space $U = \{x_1, \ldots, x_z\}$). Over a universe $U$, one can define a language $L$ as a pair $(\Sigma_L, \mathcal{I}_L)$, where $\Sigma_L$ is a finite collection of short strings over some alphabet $A$, with $|\Sigma_L| \gg |A|$, and $\mathcal{I}_L$ is a binary function $\mathcal{I}_L : U \times \Sigma_L \to \{0, 1\}$, which we call *interpreter*. We say that a phenomenon $P_i$ is *explainable* in a language $L$ if there exists a string $e \in \Sigma_L$ such that, for any $x \in U$, it occurs $\mathcal{I}_L(x, e) = \mathbf{1}_{P_i}(x)$, where $\mathbf{1}_{P_i}(x)$ is the indicator function of $P_i$. We call the string $e$ an explanation, in the language $L$, for the phenomenon $P_i$.

Our first contribution is the introduction of a new class of machine learning problems, which we refer to as *Explanatory Learning* (EL).

Consider the general problem of making a new prediction for a phenomenon $P_0 \subset U$. In our setting, this is phrased as a binary classification task: given a sample $x' \in U$, establish whether $x' \in P_0$ or not. We are interested in two instances of this problem, with different underlying assumptions:

- **The communication problem: we have an explanation**. We are given an explanation $e_0$ for $P_0$, in an unknown language $L$. This means that we do not have access

---

[1]The explanation *Four wandering stars having their period around a principal star* is adapted from the English translation of the Sidereus Nuncius (Galilei, 2016, page 9). First sketch on the left is compatible with the rule since the fourth moon can be hidden by one of the other moons or by Jupyter itself.

to an interpreter $\mathcal{I}_L$; it looks like Japanese to a non-Japanese speaker. Instead, we are also given other explanations $\{e_1, \ldots, e_n\}$, in the same language, for other phenomena $P_1, \ldots, P_n$, as well as observations of them, i.e., datasets $\{D_1, \ldots, D_n\}$ in the form $D_i = \{(x_1, \mathbf{1}_{P_i}(x_1)), \ldots, (x_m, \mathbf{1}_{P_i}(x_m))\}$, with $m \ll |U|$. Intuitively, here we expect the learner to use the explanations paired with the observations to build an approximated interpreter $\hat{\mathcal{I}}_L$, and then use it to make the proper prediction for $x'$ by evaluating $\hat{\mathcal{I}}_L(x', e_0)$.

- **The scientist problem: we do not have an explanation**. We are given explanations $\{e_1, \ldots, e_n\}$ in an unknown language $L$ for other phenomena $P_1, \ldots, P_n$ and observations of them $\{D_1, \ldots, D_n\}$. However, we do not have an explanation for $P_0$; instead, we are given just a small set of observations $D_0 = \{(x_1, \mathbf{1}_{P_0}(x_1)), \ldots, (x_k, \mathbf{1}_{P_0}(x_k))\}$ and two guarantees, namely that $P_0$ is explainable in $L$, and that $D_0$ is *representative* for $P_0$ in $L$. That is, for every phenomenon $P \neq P_0$ explainable in $L$ there should exist at least a $x_i \in D_0$ such that $\mathbf{1}_{P_0}(x_i) \neq \mathbf{1}_P(x_i)$. Again, we expect the learner to build the interpreter $\hat{\mathcal{I}}_L$, which should first guide the search for the missing explanation $e_0$ based on the clues $D_0$, and then provide the final prediction through $\hat{\mathcal{I}}_L(x', e_0)$.

Several existing works fall within the formalization above. The seminal work of Angluin (1987) on learning regular sets is an instance of the scientist problem, where finite automata take the role of explanations, while regular sets are the phenomena. More recently, CLEVR (Johnson et al., 2017) posed a communication problem in a universe of images of simple solids, where explanations are textual and read like *"There is a sphere with the same size as the metal cube"*. Another recent example is CLIP (Radford et al., 2021), where 400,000,000 captioned internet images are arranged in a communication problem to train an interpreter, thereby elevating captions to the status of explanations rather than treating them as simple labels[2]. With EL, we aim to offer a unified perspective on these works, making explicit the core problem of learning an interpreter purely from observations.

**Relationship with other ML problems.** We briefly discuss the relationship between EL and other problems in ML, pointing to Sec. 6 for additional discussion on the related work.

EL can be framed in the general meta-learning framework, where the learner gains experience over multiple tasks to improve its general learning algorithm, thus requiring less data and computation on new tasks. Yet, differently from current meta-learning approaches (Hospedales et al., 2020), we are not optimizing an explicit meta-objective. Instead, we expect the sought generality to be a consequence of learning a language from examples, rather than a goal to optimize for. EL also shares some aspects with multitask learning and few-shot learning. The core difference here resides in the additional assumption of a common language between tasks and the centrality of the interpreter learning problem. The comparison between CRN and EMP-R in the experiments shows that this additional assumption grants a disruptive increase in generalization performance over a naive multitask learning approach. Also the problem of cross-task generalization in NLP shares many aspects with the communication and scientist problem of EL, such as the common language in which the different tasks are framed, see for instance (Ye et al., 2021). Yet, it exhibits also a crucial difference: in NLP this problem is always in a text-to-text format, i.e. in NLP the problem is framed in a single textual domain where explanations and data share the same role.

To many, the concept of explanation may sound close to the concept of program; similarly, the scientist problem may seem a rephrasing of the fundamental problem of Inductive Logic Programming (ILP) (Shapiro, 1981) or Program Synthesis (PS) (Balog et al., 2017). This is not the case. ILP has the analogous goal of producing a hypothesis from positive/negative examples accompanied by background knowledge. Yet, ILP requires data expressed as logic formulas by a human expert; the ILP solver outputs a logic proposition, which the expert in turn interprets[3]. With EL, data can be fed as-is without being translated into logic propositions, and an interpreter plays the expert's role. PS also admits raw data as input, it yields a program as output, and replaces the expert with a hand-crafted interpreter; this way, the sequence of symbols produced by a PS system only makes sense to a human, not to the system itself. Instead, in EL, the interpreter is learned from scratch rather than hardcoded. An empirical comparison demonstrating the benefits of EL over PS is given in Sec. 5.

Next we introduce Odeen, an environment and benchmark to experiment with the EL paradigm.

---

[2]This shift greatly improved the performance of their model, as discussed in (Radford et al., 2021, Sec. 2.3).
[3]In this sense, ILP suffers the same problem of Explanation-Based Learning, see Minton (1990)

## 3 ODEEN: A PUZZLE GAME AS EXPLANATORY LEARNING ENVIRONMENT

**Single game.** The inset shows a typical situation in a game of Odeen. The player looks at a set of structures made of simple geometric figures. Each structure is tagged red or green according to a secret rule, and the player's goal is to guess this rule. In the example, the rule can not possibly be "A structure must contain at least one red square" since

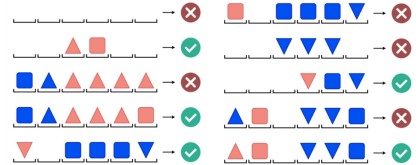

the fifth structure on the left does not contain a red square, but respects the rule (green tag). Once the player figures out the rule, she uses it to tag a large number of new structures [4]. We made a simplified interactive version of Odeen at https://bit.ly/3FeQjH0.

**Odeen challenge.** Our challenge goes beyond attempting to solve one single game, which is already a difficult problem per se. Specifically, consider the point of view of someone who does *not* speak the language in which the rules are written; an example of this is in the inset, where the secret explanations are given in hieroglyphics rather than English. Such a player would not be able to tag any structure according to the secret rule, even if the latter is given. However, assume the player has been watching several games together with their secret rules. Reasonably, the player will grow an idea of what those strange symbols mean. If the player then wins several Odeen games, it would be strong evidence of mastering the Odeen language.

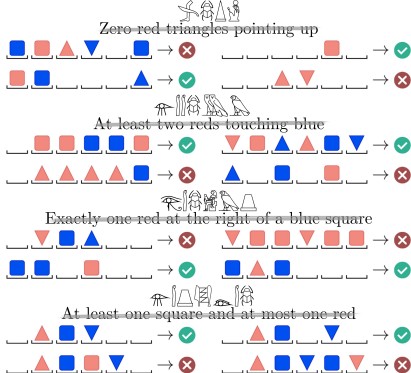

**Problem formulation.** Each game of Odeen is a different phenomenon $P_i$ of a universe $U$ whose elements $x$ are sequences of simple geometric figures. In this universe, players are scientists who try to explain the new phenomenon; see Figure 1. The specific task is to make correct predictions for a new phenomenon $P_0$ (a new game) given: (i) a few observations $D_0$ of $P_0$ (tagged structures), in conjunction with (ii) explanations $\{e_1, \ldots, e_n\}$ and observations $\{D_1, \ldots, D_n\}$ of other phenomena (other games and their secret rules); see Figure 2 (A and B). More formally:

> Let us be given $s$ unexplained phenomena with $k$ observations each, and $n$ explained phenomena with $m$ observations each; let the $n$ phenomena be explained in an unknown language, i.e., $e_1, \ldots e_n$ are plain strings without any interpreter. The task is to make $\ell$ correct predictions for each of the $s$ unexplained phenomena.

We consider $\ell = 1176$ (1% of structures); $s = 1132$; $k = 32$; $m = 10K, 1K, 100$; $n = 1438$ or $500$.

**Why not explicitly ask for the rule?** Instead of requiring the player to reveal the secret explanation explicitly, we follow the principle of zero-knowledge proofs (Blum et al., 1988). In our setting, this is done by asking the player to correctly tag many unseen structures according to the discovered rule. This makes it possible for any binary classification method to fit our EL environment without generating text. A winning condition is then defined by counting the correct predictions, instead of a textual similarity between predicted and correct explanation, which would require the player to guess word-by-word the secret rule. In fact, different phrasings with the same meaning should grant a victory, e.g., "at least one pyramid pointing up and at most one pyramid pointing up" is a winning guess for the secret rule "exactly one pyramid pointing up". A brute-force enumeration of all equivalent phrasings, in turn, would not allow solutions like "exactly one *one* pyramid pointing up", where "one" is mistakenly repeated twice; intuitively, we want to accept this as correct and dismiss the grammatical error. Similarly, a solution like "exactly one pointing up", where "pyramid" is omitted, should be accepted in a universe where only pyramids point up. We will reencounter these examples in Sec. 5 when we discuss the key properties of our approach.

**Dataset generation.** Odeen structures are sequences of up to six elements including spaces, blues or reds, squares or pyramids, the latter pointing up or down. The size of the universe is $|U| =$

---

[4]*The solution of the inset game is at the end of this footnote.* Odeen is inspired by the board game Zendo, where players must explicitly guess the rule, known only to a master. In Zendo, players can also experiment by submitting new structures to the master. *Solution: At least one square at the right of a red pyramid.*

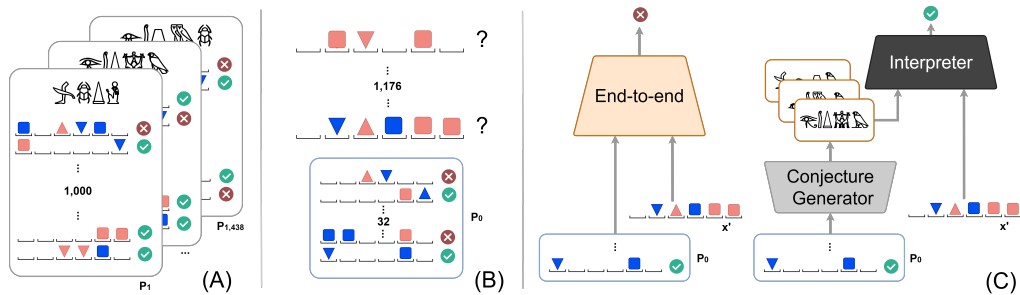

Figure 2: **Odeen Explanatory Learning problem.** Given observations and explanations in an unknown language for some phenomena (**A**), plus a few observations of a new phenomenon, explain the latter and prove this knowledge by correctly tagging a large set of new samples (**B**). An empiricist approach attempts to extract this knowledge from data (**C**, left); a rationalist one conceives data as theory-laden observations, used to find the true explanation among a set of conjectures (**C**, right).

$7^6 = 117,649$ possible structures. We further created a small language with objects, attributes, quantifiers, logical conjunctions, and interactions (e.g., "touching", see Appendix A). The grammar generates $\approx$25k valid rules in total. Each of the $|U|$ structures is tagged according to all the rules. The tagging is done by an interpreter implemented via regular expressions.

**Metrics.** As described above, the task is to tag $\ell$ new structures for each of $s$ unexplained games. An EL algorithm addressing this task encodes the predicted rule as an $\ell$-dimensional binary vector $\mathbf{v}$ per game (predicted vector), where $v_i = 1$ means that the $i$-th structure satisfies the predicted rule, and $v_i = 0$ otherwise (see inset). Let $\mathbf{w}^*$ be the ground-truth vector, obtained by tagging the $\ell$ structures according to the correct secret rule. Then, the Hamming distance $d_H(\mathbf{v}, \mathbf{w}^*)$ measures the number of wrong tags assigned by the EL algorithm; if $d_H(\mathbf{v}, \mathbf{w}^*) < d_H(\mathbf{v}, \mathbf{w}_i)$, where $\mathbf{w}_i \neq \mathbf{w}^*$ ranges over all the possible $\approx$25k rules, then the predicted rule $\mathbf{v}$ made by the algorithm is deemed correct.

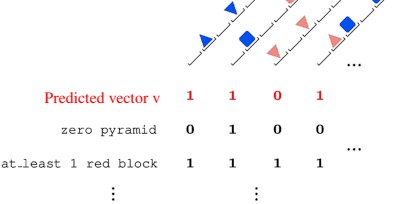

According to this, the *Nearest Rule Score* (NRS) is the number of correctly predicted rules over a total of $s$ games. A second score, the *Tagging Accuracy* (T-Acc), directly counts the number of correct tags averaged over $s$ games; this is more permissive in the following sense. Consider two different rules $A$ and $B$ sharing 99% of the taggings, and let $A$ be the correct one; if an EL model tags all the structures according to the *wrong* rule $B$, it still reaches a T-Acc of 99%, but the NRS would be 0. An EL algorithm with these scores would be good at making predictions, but would be based on a wrong explanation.

## 4 CRITICAL RATIONALIST NETWORKS

In principle, an EL problem like Odeen can be approached by training an end-to-end neural network to predict $\hat{y} = \mathbf{1}_{P_i}(x')$, given as input a set of observations $D_i$ and a single sample $x'$ (see Figure 2 C). Such a model would assume that all the information needed to solve the task is embedded in the data, ignoring the explanations; we may call it a "radical empiricist" approach (Pearl, 2021). A variant that includes the explanations in the pipeline can be done by adding a textual head to the network. This way, we expect performance to improve because predicting the explanation string can aid the classification task. As we show in the experiments, the latter approach (called "conscious empiricist") indeed improves upon the former; yet, it treats the explanations as mere data, nothing more than mute strings to match, in a Chinese room fashion (Searle, 1980; Bender & Koller, 2020).

In the following, we introduce a "rationalist" approach to solve EL problems. This approach recognizes the given explanations as existing knowledge, and focuses on interpreting them. Here theory comes first, while the data become theory-laden observations.

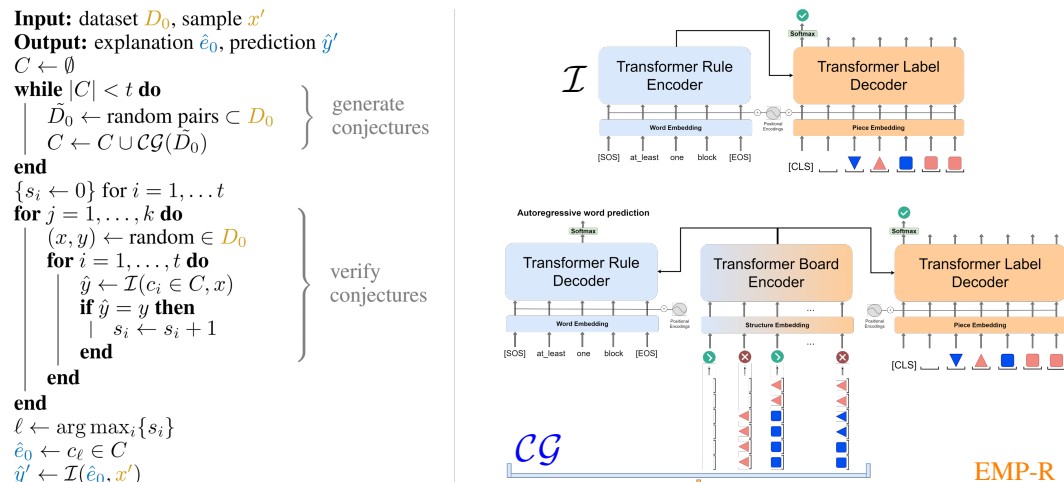

**Input:** dataset $D_0$, sample $x'$
**Output:** explanation $\hat{e}_0$, prediction $\hat{y}'$
$C \leftarrow \emptyset$
**while** $|C| < t$ **do**
    $\tilde{D}_0 \leftarrow$ random pairs $\subset D_0$
    $C \leftarrow C \cup \mathcal{CG}(\tilde{D}_0)$    } generate conjectures
**end**
$\{s_i \leftarrow 0\}$ for $i = 1, \ldots t$
**for** $j = 1, \ldots, k$ **do**
    $(x, y) \leftarrow$ random $\in D_0$
    **for** $i = 1, \ldots, t$ **do**
        $\hat{y} \leftarrow \mathcal{I}(c_i \in C, x)$  } verify conjectures
        **if** $\hat{y} = y$ **then**
            $s_i \leftarrow s_i + 1$
        **end**
    **end**
**end**
$\ell \leftarrow \arg\max_i \{s_i\}$
$\hat{e}_0 \leftarrow c_\ell \in C$
$\hat{y}' \leftarrow \mathcal{I}(\hat{e}_0, x')$

Figure 3: **Left:** Test-time algorithm of CRNs. **Right:** CRNs are implemented using encoder-decoder transformers blocks, details of the parameters in Appendix B. **Right-top:** $\mathcal{I}$ denotes the interpreter model (rule encoder and label decoder). **Right-bottom:** The conjecture generator $\mathcal{CG}$ is composed by blue blocks. The "radical empiricist" (EMP-R) is composed by orange blocks. The "conscious empiricist" (EMP-C) baseline model consists of all the transformer blocks in the right-bottom figure, board encoder with rule and label decoders (all the blue and orange blocks).

**Learning model.** Our *Critical Rationalist Networks* (CRNs) tackle the EL scientist problem introduced in Sec. 2: to find $y = \mathbf{1}_{P_0}(x')$ given $x'$, $D_0$, $\{D_1, \ldots, D_n\}$, $\{e_1, \ldots, e_n\}$. They are formed by two independently trained models:

(i) A stochastic *Conjecture Generator*

$$\mathcal{CG} : \{(x, \mathbf{1}_P(x))_j\}_{j=1}^k \mapsto e,$$

taking $k \leq |D_0|$ pairs $(x, \mathbf{1}_P(x)) \in D_i$ as input, and returning an explanation string $e \in \Sigma$ as output. $\mathcal{CG}$ is trained to maximize the probability that $\mathcal{CG}(\tilde{D}_i) = e_i$ for all $i = 1, \ldots, n$, where $\tilde{D}_i \subset D_i$ is a random sampling of $D_i$, and $|\tilde{D}_i| = k$.

(ii) A learned *Interpreter*

$$\mathcal{I} : (e, x) \mapsto \hat{y},$$

which takes as input a string $e \in \Sigma$ and a sample $x \in U$, to output a prediction $\hat{y} \in \{0, 1\}$. $\mathcal{I}$ is trained to maximize the probability that $\mathcal{I}(e_i, x) = \mathbf{1}_{P_i}(x)$, with $i = 1, \ldots, n$ and $(x, \mathbf{1}_{P_i}(x)) \in D_i$.

At test time, we are given a trained $\mathcal{CG}$ and a trained $\mathcal{I}$, and we must predict whether some $x' \notin D_0$ belongs to $P_0$ or not. The idea is to first generate $t$ conjectures by applying $\mathcal{CG}$ $t$ times to the dataset $D_0$; then, each conjecture is verified by counting how many times the interpreter $\mathcal{I}$ outputs a correct prediction over $D_0$. The conjecture with the highest hit rate is our candidate explanation $\hat{e}_0$ for $P_0$. Finally, we obtain the prediction $\hat{y}'$ as $\mathcal{I}(\hat{e}_0, x')$. See Figure 3 (left) for a step-by-step pseudo code.

**Remarks.** The interpreter $\mathcal{I}$ is a crucial component of our approach. A poor $\mathcal{I}$ may fail to identify $e_0$ among the generated conjectures, or yield a wrong prediction $y'$ when given the correct $e_0$. On the other hand, we can work with a $\mathcal{CG}$ of any quality and safely return as output an *unknown* token, rather than a wrong prediction, whenever $e_0$ does not appear among the generated conjectures. The role of $\mathcal{CG}$ is to trade-off performance for computational cost, and is controlled by the parameter $t$. Larger values for $t$ imply more generated conjectures, corresponding to exhaustive search if taken to the limit (as done, e.g., in Radford et al. (2021)). This potential asymmetry in quality between $\mathcal{CG}$ and $\mathcal{I}$ is tolerated, since the learning problem solved by $\mathcal{CG}$ is generally harder.

Secondly, although a CRN is implemented using neural networks, as we shall see shortly, its working hypothesis does not coincide with a snapshot of the countless network's parameters; rather, the working hypothesis is but a small conjecture analyzed at a given moment. This way, the CRN hypothesis is detached from the model and can only be accepted or refused in its entirety, rather than being slightly adjusted at each new data sample (Figure 2 C, the hypothesis is in orange).

**Implementation.** Figure 3 (right) illustrates the architecture of CRNs, which we implement using encoder-decoder transformers (Vaswani et al., 2017). The figure also shows the architecture of the baseline methods EMP-R and EMP-C, corresponding to the end-to-end NN model and its variant with a textual head, respectively. We refer to the Appendix for further details.

## 5 EXPERIMENTS

We extensively compared CRNs to the radical (EMP-R) and conscious (EMP-C) empiricist models over the Odeen challenge, and analyzed several fundamental aspects.

**Generalization power.** The Odeen challenge directly addresses the generalization capability of a given algorithm, by asking for explanations to unexplained phenomena. This is evaluated over $s = 1132$ new games, where each game is given with $k = 32$ tagged structures (guaranteed to satisfy a unique, yet unknown rule) and requires to correctly tag $\ell = 1176$ unseen structures according to the unknown rule. The training set are $n = 1438$ games with ground-truth explanations and $m = 1000$ tagged structures per game. The test set does not include *any* rule equivalent to the training rules. One important example is the bigram "exactly two", which appears in the test set, but was deliberately excluded from training; the training rules only contain "at least/most two" and "exactly one". The CRN guessed $40\%$ of the 72 test rules with "exactly two", while the empiricist models (EMP-C, EMP-R) scored $4\%$ and $0\%$ respectively. The table below reports the full results.

The NRS of $77.7\%$ denotes that the CRN discovered the correct explanation for 880 out of 1132 new phenomena. Using the same data and a similar number of learnable parameters, the empiricist models score $22.5\%$ at most. Some example games can be found in Appendix D.

| MODEL | NRS | T-ACC | R-ACC |
|---|---|---|---|
| CRN | **0.777** | **0.980** | **0.737** |
| EMP-C | 0.225 | 0.905 | 0.035 |
| EMP-R | 0.156 | 0.898 | - |

In the table, R-Acc measures how frequently an output explanation is equivalent to the correct one; two rules $A$ and $B$ are equivalent if the tags assigned by the hard-coded interpreter to all the $\sim$117k structures in $U$ are the same for $A$ and $B$.

As expected, the explanation predicted by the conscious empiricist model is rarely correct (R-Acc $3.5\%$), even when it tags some structures properly (NRS $22.5\%$); indeed, EMP-C gives no guarantee for the predicted explanation to be consistent with the tags prediction. Conversely, the CRN consistently provides the correct explanation when it is able to properly tag the new structures (NRS $77.7\%$, R-Acc $73.7\%$). The $4\%$ gap between the two scores is clarified in the next paragraph.

**Handling ambiguity and contradiction.** One may reasonably expect that a CRN equipped with the ground-truth interpreter used to generate the dataset, would perform better than a CRN with a learned interpreter. Remarkably, this is not always the case, as reported in Table 1.

Table 1: **Explanatory Learning vs Program Synthesis paradigm.** Performance comparison of a data-driven vs ground-truth interpreter in a CRN. The last column shows the tag prediction accuracy of the learned $\mathcal{I}$, when provided with the correct rule.

| | | NRS | | T-ACC |
|---|---|---|---|---|
| TRAIN DATA | | FULLY-LEARNED CRN | HARDCODED $\mathcal{I}$ CRN | LEARNED $\mathcal{I}$ |
| 10K STRUCT. | 1438 RULES | **0.813** | 0.801 | 0.997 |
| 1K STRUCT. | 1438 RULES | **0.777** | 0.754 | 1.000 |
| 100 STRUCT. | 1438 RULES | 0.402 | **0.406** | 0.987 |
| 10K STRUCT. | 500 RULES | 0.354 | **0.377** | 0.923 |
| 1K STRUCT. | 500 RULES | 0.319 | **0.336** | 0.924 |
| 100 STRUCT. | 500 RULES | **0.109** | 0.101 | 0.920 |

The better performance of the fully learned interpreter over the ground-truth one is due to its ability to process ill-formed conjectures generated by the $\mathcal{CG}$. The conjecture "at least one pointing up" makes the hard-coded interpreter fail, since "pointing up" must always follow the word "pyramid" by the grammar. Yet, in Odeen, pyramids are the only objects that point, and the learned $\mathcal{I}$ interprets

the conjecture correctly. Other examples include: "exactly one red block touching pyramid blue" ("pyramid" and "blue" are swapped), or the contradictory "at least one two pyramid pointing up and exactly one red pyramid", which was interpreted correctly by ignoring the first "one". When the learned interpreter is not very accurate, the negative effect of errors in tagging prevails.

Making sense out of ambiguous or contradictory messages[5] is a crucial difference between a learned interpreter vs a hardcoded one. As Rota (1991) reminds us, a concept does not need to be precisely defined in order to be meaningful. Our everyday reasoning is not precise, yet it is effective. "After the small tower, turn right"; we will probably reach our destination, even when our best attempts at defining "tower", as found, e.g., in the Cambridge dictionary, begin with "a *tall*, narrow structure...".

**Explainability.** The predictions of a CRN are *directly caused* by a human-understandable explanation that is available in the output; this makes CRNs explainable by construction. Further, CRNs allow counterfactuals; one may deliberately change the output explanation with a new one to obtain a new prediction. The bank ML algorithm spoke: "Loan denied"; explanation: "Two not paid loan in the past and resident in a district with a high rate of insolvents". With a CRN, we can easily discard this explanation and compute a new prediction for just "Two not paid loan in the past". Importantly, by choosing a training set, we control the language used for explanations. This allows a CRN to ignore undesirable patterns in the data (e.g., skin color) if these can not be expressed in the chosen language. If the Odeen training set had no rule with "pointing up/down", the learned interpreter would see all equal pyramids, even with unbalanced training data where 90% of pyramids point up.

On the contrary, current explainability approaches for NNs (end-to-end empiricist models) either require some form of reverse engineering, e.g., by making sense out of neuron activations (Goh et al., 2021), or introduce an ad-hoc block to generate an explanation given a prediction, without establishing a cause-effect link between the two (Hendricks et al., 2016; Hind et al., 2019).

**Adjustable thinking time.** End-to-end models do not exhibit a parameter to adjust their processing to the complexity of the incoming prediction. By contrast, CRNs have a test-time parameter $t$, corresponding to the number of generated conjectures, which trades off computational cost for performance. In the inset, we plot the cumulative R-Acc score ($y$ axis) against the number $t$ of generated conjectures ($x$ axis). The curves show that $> 60\%$ of correct explanations are found within the first 50 candidates, and $> 80\%$ are within the first 300. As a reference, a brute

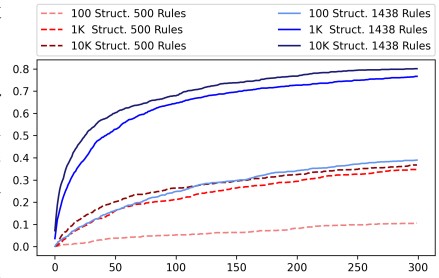

force exhaustive search would reach 100% over a search space of $24,794$ possible explanations.

**Prediction confidence.** As explained in Sec. 4, at test time the CRN selects the conjecture with the highest hit rate among the ones generated by the $\mathcal{CG}$. Alternatively, one may keep only the conjectures coherent with *all* the structures in the table, returning an "unknown explanation" signal if no such conjectures are found. If the interpreter is sufficiently accurate, this stricter condition barely deteriorates the CRN performance (1-2% gap on average), and it will never return a prediction based on a possibly wrong explanation. For example, tested in a setting with $n = 1438$, $m = 1000$ (same as the second row of Table 1), this stricter CRN discovers the correct explanation for 861 out of 1132 new phenomena (76%), and admits its ignorance on the other 271. Conversely, evaluating the confidence of an end-to-end neural network remains an open problem (Meinke & Hein, 2019).

## 6 RELATED WORK

**Epistemology.** The deep learning model we propose in this work, CRNs, is designed according to the epistemological theory of critical rationalism advanced by Popper (1935), where knowledge derives primarily from conjectures, criticized at a later stage using data. Deutsch (2011) remarks that to make this critique effective, conjectures should not be adjustable but can only be kept or rejected at each new data sample, as done in CRNs at test time. Only in this way we can discover explanations with "reach", namely that maintain predictive power in novel situations.

---

[5]This is one of seven essential abilities for intelligence as found in *GEB* (Hofstadter, 1979, Introduction).

**Machine learning.** Explanatory Learning enriches the fundamental problem of modern program synthesis (e.g., Balog et al., 2017; Ellis et al., 2020) by including the interpretation step among what should be learned. As seen with a few examples (see Handling ambiguity in Sec. 5), a CRN with learned $\mathcal{I}$ can exploit the ambiguity of language to impose new meaning on arbitrary substrates, which Santoro et al. (2021) recognize as a fundamental trait of symbolic behavior. Recent literature finds few yet remarkable approaches that fit our EL paradigm, such as CLIP (Radford et al., 2021) in the vision area, and Generate & Rank (Shen et al., 2021) for Math Word Problems in NLP.

The Odeen challenge continues the tradition of AI benchmarks set in idealized domains (Mitchell, 2021). Unlike CLEVR (Johnson et al., 2017) and ShapeWorld (Kuhnle & Copestake, 2017), Odeen focuses on abduction rather than deduction. Unlike ARC (Chollet, 2019), Odeen is a closed environment providing all it takes to learn the language needed to solve it. Unlike the ShapeWorld adaptation of Andreas et al. (2017), its score is measured in terms of discovered explanations rather than sparse guessed predictions; further, the test and training set do not share any phenomenon.

**Learning theory.** Finally, we point out that the expression *Explanatory Learning* was previously used by Aaronson (2013, Sec. 7) to argue about the necessity of a learning theory that models "predictions about phenomena different in kind from anything observed". The author pointed to the work of Angluin (1987), who generalized the PAC model (Valiant, 1984) by moving the goal from successful predictions to comprehensive explanations. This setup was discussed by Valiant himself in *Robust Logics* (Valiant, 2000), and pushed forward in (Michael & Valiant, 2008; Michael, 2014), which present a dual inference scheme similar to the CRN, and in Možina et al. (2007), where data labels are motivated by explanations (arguments). In a recent work Michael (2019) introduces a variant of the typical PAC definition that conceives the dual nature of explanations that emerges in this work, as propositions of a language to learn, and as explanation of a whole phenomenon.

Finally, we want to remark that Explanatory Learning should not be confused with Explainable Learning, a branch of XAI. While the goal of EL is to achieve generality by means of language explanations, XAI focuses on the explanation itself, with the ultimate goal of making AI predictions more digestible for humans (Hind et al., 2019; Teso & Kersting, 2019).

## 7 Conclusions

Recently, the attention on the epistemological foundations of deep learning has been growing. The century-old debate between empiricists and rationalists about the source of knowledge persists, with two Turing prizes on opposite sides; LeCun (2019) argues that empiricism still offers a fruitful research agenda for deep learning, while Pearl (2021) supports a rationalist steering to embrace model-based science principles. This new debate is relevant, since as Pearl notes, today we can submit the balance between empiricism and innateness to experimental evaluation on digital machines.

**Limitations and future directions.** EL models the essential part of the knowledge acquisition process, namely the interval that turns a mute sequence of symbols into an explanation with reach. However, our modeling assumes a representative set of observations $D_0$ to be given (the $k = 32$ structures of the new phenomenon). A more comprehensive explanatory model would allow the player to do without these observations, and instead include an interaction phase with the environment where the $D_0$ itself is actively discovered. We see this as an exciting direction for follow-ups.

Odeen has potential as a parametric environment to experiment with EL approaches. The structure length (6 in this paper), the number of shapes (3), attributes (2), and the grammar specifications (see Appendix A) can be easily tweaked to obtain either simpler or significantly more complex environments. The design choices of this paper provide a good starting point; the resulting benchmark is far from being saturated, but experimenting with different variants is a possibility for the future.

Finally, we expect CRNs to be more resilient than end-to-end models to adversarial attacks. For a given data point $x' \in P_0$ classified correctly by an empiricist model, a small adversarial change on $D_0$ can flip the prediction for $x'$ while remaining unnoticed. Conversely, suppose that a CRN made the prediction for $x'$, and assume that the correct explanation was ranked as the 5th most likely by the $\mathcal{CG}$. The same attack on $D_0$ will have the effect of moving the correct explanation lower in the ranking; however, as long as it stays within the first $t$ conjectures (300 in this paper), it will always be found by the interpreter as the correct solution.

## 7.1 ETHICS STATEMENT

The Explainability paragraph in Section 5 briefly touches upon the topic of fairness in AI, pointing to a possible way to address data debiasing. To this end, our proposed CRN model can be beneficial. In particular, CRNs can be potentially used to *reduce* the data bias, which is often identified as one possible cause of algorithmic discrimination in automated decision processes. While addressing these aspects is out of scope for this paper and it certainly deserves deeper investigation, we do not foresee any potentially harmful or inappropriate application of our methodology.

## 7.2 REPRODUCIBILITY STATEMENT

**Training:** The proposed CRN model is composed of two parts, a learnable interpreter $\mathcal{I}$ and a conjecture generator $\mathcal{CG}$. Their architecture is described in Figure 3 (right) and in Appendix B. The training procedure, including the choice of hyperparameters, is also described in Appendix B. **Testing:** The algorithm used at test time is fully described in Figure 3 (left). **Data:** One of the main contributions of the paper is the introduction of a new dataset and benchmark, called Odeen. Its full description is given in Section 3 (Problem formulation and Dataset generation paragraphs) and in Appendix A. The latter section also includes a formal definition of the Odeen grammar. The metrics used for evaluation in the Odeen benchmark are defined in Section 3 (Metrics paragraph). A simple interactive version of the Odeen game, to help the readers familiarize with the concept, is available at https://bit.ly/3FeQjH0. Full code and data of the entire benchmark will be made publicly available upon acceptance.

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

## A  FURTHER DETAILS ON THE ODEEN DATASET

**Training set.**  The total number of rules produced by the Odeen grammar is 24,794. We consider training sets varying from 500 to 1438 rules. We choose these rules such that each token and each syntactic construct appears at least once; then, we uniformly select the others from the distribution. We removed from the training set any rule containing the bigram `exactly 2`, as well as any rule of the form `at_least 2 X and at_most 2 X`, equivalent to `exactly 2 X`. Each rule is associated with a set of 32 labelled structures that unambiguously identify a rule equivalence class. The first 10 structures are chosen by searching pairs of similar structures with different labels, following a common human strategy in Zendo. The remaining 22 structures are selected to ensure the lack of ambiguity on the board.

**Test set.**  We generate the 1,132 games that compose the test set the same way, with the additional constraint of excluding the rules belonging to an equivalence class that is already in the training set. In the test set, 72 rules contain the bigram `exactly 2`.

**Formal definition of the Odeen grammar.** The context-free grammar in Figure 4 defines all the acceptable rules in Odeen. This grammar only formalizes which rules are *syntactically correct*. Token names (e.g. red, 1 or touching) do not imply any rule meaning.

The hard-coded interpreter formalizes how to interpret the rules. Similarly to compilers, it tokenizes and transforms the rule into an abstract syntax tree (AST). The interpreter then adds semantic information to the AST, establishing the truth value of each node based on the truth value of its children and the structure under evaluation.

**The Odeen binary semantic representations.** By simulating the process of scientific discovery, Odeen offers a convenient simulation of a world described by a language. Besides the computational tractability, the simplicity and adjustable size of the Odeen world allows us explicit the whole semantics of its language.

These semantics can be encoded in a binary *semantic matrix* $S$ with the 24,794 rules $e_i$ on the rows and the 117,649 structures $x_j$ on the columns. The $s_{ij}$ element of this matrix is equal to 1 if the structure $x_j$ complies with the rule $e_i$ and 0 otherwise, see inset in Section 3, Metrics paragraph. $S_{i*}$, the 117,649-dimensional binary vector coinciding with the $i$-th row of $S$, fully represents the meaning of rule $e_i$ in the Odeen world. Similarly, each structure $x_j$ is represented by the 24,794-dimensional binary vector coinciding with the column $S_{*j}$ of $S$.

In Figure 5, we analyze the distribution of the Hamming weights (i.e., the number of ones) in $\{S_{i*}\}_{i=1}^{117,649}$ (5a) and $\{S_{*j}\}_{j=1}^{24,794}$ (5b). We observe an asymmetry between the rule and structure distributions. On one hand, the semantic representation of a rule can be quite unbalanced, with populated extremes of rules evaluating *all* structures with 1 (or 0) as shown in Figure 5a. On the other hand, Figure 5b shows that the semantic representations of structures are very balanced; most of them have around half zeros and half ones, with no structure with less than 10k or more than 14k ones.

This balanced trend, along with the well separable PCA of $\{S_{*j}\}_{j=1}^{24,794}$ (Figure 6a) suggests that the chosen semantics produce representations that are effective in separating structures. Conversely, the PCA of $\{S_{i*}\}_{i=1}^{117,649}$ is much less homogeneous (Figure 6b). Here we can recognize two poles, corresponding respectively to rules with all ones and all zeros. We believe that this analysis of the binary semantic representations is only partial, and we leave further exploration for follow-up work.

## B    IMPLEMENTATION DETAILS

In this paragraph, we give the implementation details of the models proposed and depicted in Figure 3 (right). All the models are based on a Transformer block composed of 4 layers and 8 heads. We used a hidden dimension of 256 for all the models except for the interpreter, where we used a hidden dimension of 128. The models differ primarily by the type of transformer block used (encoder/decoder), inputs and embeddings. In detail:

$$
\begin{aligned}
\langle\text{RULE}\rangle &\models \langle\text{PROP\_S}\rangle \mid \langle\text{PROP}\rangle \mid \langle\text{PROP\_S}\rangle \ \langle\text{CONJ}\rangle \ \langle\text{PROP\_S}\rangle \\
\langle\text{PROP}\rangle &\models \langle\text{QTY}\rangle \ \langle\text{OBJ}\rangle \ \langle\text{REL}\rangle \ \langle\text{OBJ}\rangle \\
\langle\text{PROP\_S}\rangle &\models \langle\text{QTY}\rangle \ \langle\text{OBJ}\rangle \\
\langle\text{OBJ}\rangle &\models \langle\text{COL}\rangle \mid \langle\text{SHAPE}\rangle \mid \langle\text{COL}\rangle \ \langle\text{SHAPE}\rangle \\
\langle\text{QTY}\rangle &\models \texttt{at\_least} \ \langle\text{NUM}\rangle \mid \texttt{exactly} \ \langle\text{NUM}\rangle \mid \texttt{at\_most} \ \langle\text{NUM}\rangle \mid \texttt{zero} \\
\langle\text{SHAPE}\rangle &\models \texttt{pyramid} \ \langle\text{ORIEN}\rangle \mid \texttt{pyramid} \mid \texttt{block} \\
\langle\text{REL}\rangle &\models \texttt{touching} \mid \texttt{surrounded\_by} \mid \texttt{at\_the\_right\_of} \\
\langle\text{ORIEN}\rangle &\models \texttt{pointing\_up} \mid \texttt{pointing\_down} \\
\langle\text{NUM}\rangle &\models \texttt{1} \mid \texttt{2} \\
\langle\text{CONJ}\rangle &\models \texttt{and} \mid \texttt{or} \\
\langle\text{COL}\rangle &\models \texttt{red} \mid \texttt{blue}
\end{aligned}
$$

Figure 4: Grammar productions for the Odeen Language.

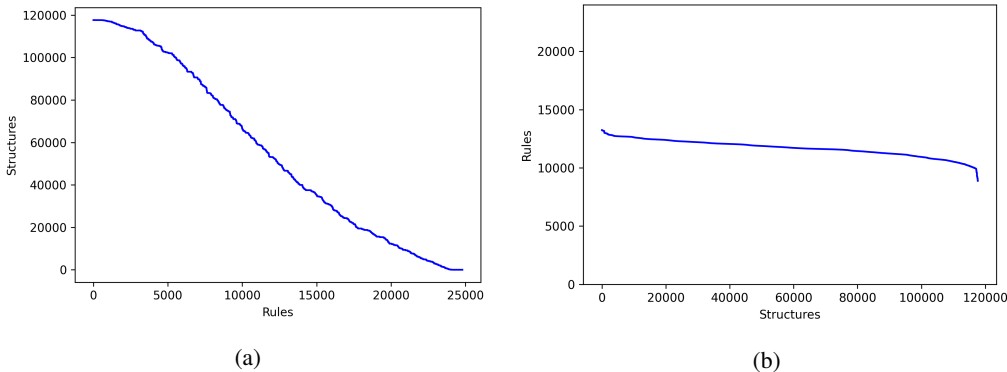

(a)                                                    (b)

Figure 5: Hamming weight of the binary semantic representation of each rule (a) and each structure (b). We sort them in descending order for visualization purposes.

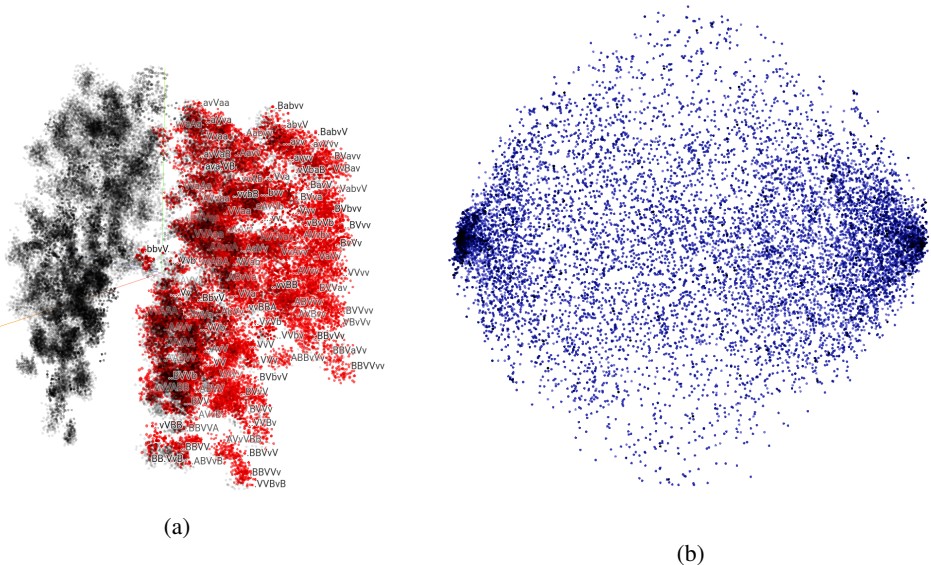

(a)

(b)

Figure 6: PCA applied to the binary semantic representation of structures (a) and rules (b). We highlight in red the structures that have two touching pyramids pointing down. To ease the visualization, we pair every structure with a different string of characters. Each character replaces an element of Odeen according to a well-defined mapping. The rules distribution reflect what can be observed in Figure 5a.

- TRANSFORMER LABEL DECODER. This is a transformer block used to predict a label given a structure. The input structure is a sequence of six learned embeddings, one per piece. We add a sinusoidal positional encoding to each embedding as in the original transformer implementation. The embedding size is 128 in the $\mathcal{I}$ and 256 in the Empiricist models (EMP-C, EMP-R). We used the standard transformer encoder block and added a special token [CLS] at the beginning of the structure like in Devlin et al. (2019) to perform the classification task.

- TRANSFORMER RULE DECODER. This is a transformer decoder block with embedding size of 128 and sinusoidal positional encoding. This decoder block is used to generate the rule by the EMP-C and $\mathcal{CG}$ models.

- TRANSFORMER BOARD ENCODER. This is a transformer encoder block used to encode the (structure, label) pairs. The input is encoded a sequence of 32 learned embeddings, one per structure-label pair. The size of each embedding is 256. We did not add positional

encodings, since the specific position of structure-label pairs among the 32 is not relevant. This block is used in all the models.

- TRANSFORMER RULE ENCODER. This transformer encoder block is used in $\mathcal{I}$ to encode the rule. Its implementation is analogous to the TRANSFORMER RULE DECODER, with the only difference that it does not use causal attention since it is an encoder layer.

Table 2: Number of training epochs for each training regimen.

| TRAINING REGIMEN | | NUMBER OF EPOCHS |
|---|---|---|
| 10K STRUCT. | 1438 RULES | 2 |
| 1K STRUCT. | 1438 RULES | 20 |
| 100 STRUCT. | 1438 RULES | 200 |
| 10K STRUCT. | 500 RULES | 6 |
| 1K STRUCT. | 500 RULES | 58 |
| 100 STRUCT. | 500 RULES | 576 |

**Training Procedure.** All the models are trained with a learning rate of $3 \cdot 10^{-4}$ using Adam (Kingma & Ba, 2017), a batch size of 512 and early-stop and dropout set to 0.1 to prevent overfitting. We train all the models on randomly sampled sets of 32 (structure, label) pairs to prevent overfitting on specific boards. Table 2 describes the number of epochs for each training regimen. Models are trained to: predict the label of a structure give the board (EMP-R); predict the label of a structure given the 32 pairs (structure, label) and the associated rule (EMP-C); predict the rule given the 32 pairs ($\mathcal{CG}$); predict the label of a structure given a rule ($\mathcal{I}$).

## C EFFICIENCY

**Data efficiency** In the Odeen challenge, CRNs require less training data to match the performance of empiricist models. For instance, in the case of 1438 rules at training, we see in Table 3 that the CRN trained 100 structures per rule (NRS$= 40, 2\%$) still overcomes the performance of empiricist models trained on a dataset 100 times bigger (NRS$= 35.2\%$ on 10k structures per rule).

**Computational cost.** In this section, we discuss the computational cost at test time of the rationalist and empiricists approaches. We discuss the results reported in Tables 4 and 5 respectively for tagging $s$ new structures and for explicitly predicting the textual rule from the board. We evaluate the cost per rule in two ways: i) by counting the number of calls of each trained neural network and ii) by measuring the absolute time in seconds of each method with the same hardware configuration.

We refer to the first quantity as the *Computational Cost* and parametrize it in terms of the main blocks of the models. This value is independent of the batch size and the hardware adopted. As an example, the cost of tagging the new structures for a CRN using 300 conjectures is given by:

Table 3: T-Acc and NRS for different training regimens.

| TRAIN DATA | MODEL | NRS | T-ACC | TRAIN DATA | MODEL | NRS | T-ACC |
|---|---|---|---|---|---|---|---|
| 10K STRUCT. 1438 RULES | CRN | **0.813** | **0.984** | 10K STRUCT. 500 RULES | CRN | **0.354** | **0.932** |
| | EMP-C | 0.352 | 0.930 | | EMP-C | 0.095 | 0.869 |
| | EMP-R | 0.179 | 0.895 | | EMP-R | 0.068 | 0.863 |
| 1K STRUCT. 1438 RULES | CRN | **0.777** | **0.980** | 1K STRUCT. 500 RULES | CRN | **0.319** | **0.930** |
| | EMP-C | 0.225 | 0.905 | | EMP-C | 0.088 | 0.874 |
| | EMP-R | 0.156 | 0.898 | | EMP-R | 0.084 | 0.876 |
| 100 STRUCT. 1438 RULES | CRN | **0.402** | **0.939** | 100 STRUCT. 500 RULES | CRN | 0.109 | **0.883** |
| | EMP-C | 0.125 | 0.865 | | EMP-C | 0.057 | 0.823 |
| | EMP-R | 0.163 | 0.896 | | EMP-R | **0.117** | 0.872 |

$$300 \cdot \mathcal{CG} + 300 \cdot b \cdot \mathcal{I} + s \cdot \mathcal{I}.$$

Where $300 \cdot \mathcal{CG}$ stands for the 300 beams used to get 300 conjectures from the conjecture generator $\mathcal{CG}$. Each conjecture (300) is then tested on all the board structures ($b$) by the interpreter $\mathcal{I}$. Finally $\mathcal{I}$ is called to apply the chosen conjecture on each new structure ($s$). As an upper bound, an exhaustive search algorithm (Exv src) uses no conjecture generator, and thus has to evaluate all admissible rules ($r$) on each structure on the board with $\mathcal{I}$. Conversely, the empiricist approach provide label predictions through a single end-to-end model which is simply called $s$ times. Concerning the problem of inferring explicitly the textual rule, using more beams in the empiricists models does not provide any increase in performance, i.e. the true rule is not a more probable proposition accessible through a larger beam search.

We measured also the absolute time in seconds with the following hardware configuration for all the experiments: 1 single core hyper threaded Xeon CPU Processor with 2.2 Ghz, 2 threads; 12.7 GiB. of RAM; a Tesla T4 GPU, with 320 Turing Tensor Core, 2,560 NVIDIA CUDA cores, and 15.7 GDDR6 GiB of VRAM.

Table 4: Computational cost of our models at test time to tag $s$ new structures. In Odeen $r$=24,794, $b$=32, $s$=1,176. Notice how CRNs offer a good balance between computational efficiency and performance, this trade-off is regulated by a single parameter, the number of beams.

| MODEL | COMPUTATIONAL COST | T (S) | NRS |
|---|---|---|---|
| EXV SRC | $r \cdot b \cdot \mathcal{I} + s \cdot \mathcal{I}$ | 47.9 | 0.99 |
| CRN [300B] | $300 \cdot \mathcal{CG} + 300 \cdot b \cdot \mathcal{I} + s \cdot \mathcal{I}$ | 0.79 | 0.81 |
| CRN [10B] | $10 \cdot \mathcal{CG} + 10 \cdot b \cdot \mathcal{I} + s \cdot \mathcal{I}$ | 0.43 | 0.35 |
| EMP | $s \cdot$ EMP-R | 0.15 | 0.35 |

Table 5: Computational cost of our models at test time to produce the textual rule in output

| MODEL | COMPUTATIONAL COST | T (S) | R-ACC |
|---|---|---|---|
| EXV SRC | $r \cdot b \cdot \mathcal{I}$ | 47.8 | 0.99 |
| CRN [300B] | $300 \cdot \mathcal{CG} + 300 \cdot b \cdot \mathcal{I}$ | 0.72 | 0.77 |
| CRN [10B] | $10 \cdot \mathcal{CG} + 10 \cdot b \cdot \mathcal{I}$ | 0.35 | 0.35 |
| EMP [300B] | $300 \cdot$ EMP-C | 0.41 | 0.07 |
| EMP [10B] | $10 \cdot$ EMP-C | 0.10 | 0.07 |
| EMP [1B] | $1 \cdot$ EMP-C | 0.10 | 0.07 |

# D  ODEEN EXAMPLE GAMES

In this section we propose a collection of qualitative results showing a series of Odeen games from the test set and how they are solved by the proposed models. For each model, we report the predicted rule (if output by the model), the accuracy on the structures labeling (T-acc), and a mark that indicates whether the nearest rule is the correct one (NRS). All the models are trained on 10000 structures with 1438 rules.

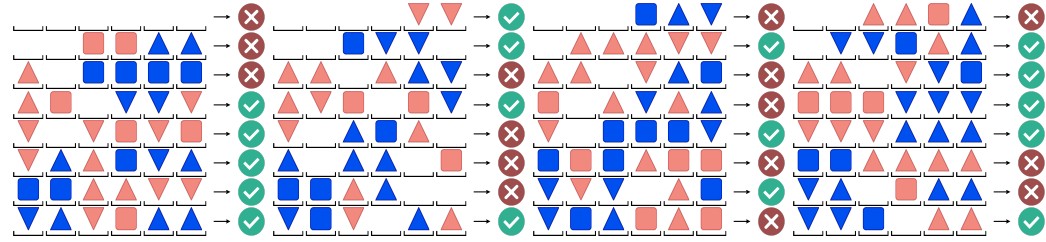

Board 01
Golden Rule: "at_least 2 pyramid pointing_down"
**CRN**: "at_least 2 pyramid pointing_down"; T-acc 1.0 ✓
**EMP-C**: "at_least 1 pyramid touching touching"; T-acc: 0.76 ✗
**EMP-R**: T-acc 0.72 ✗

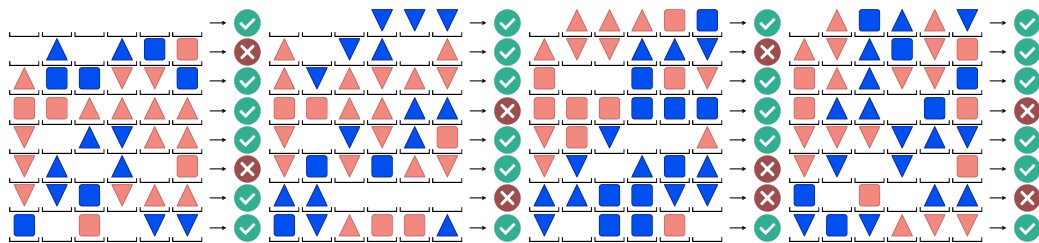

Board 04
Golden Rule: "at_most 1 blue pyramid pointing_up"
**CRN**: "zero blue or at_most 1 blue pyramid pointing_up"; T-acc 1.0 ✓
**EMP-C**: "zero 1 blue touching or or"; T-acc: 0.89 ✗
**EMP-R**: T-acc 0.92 ✓

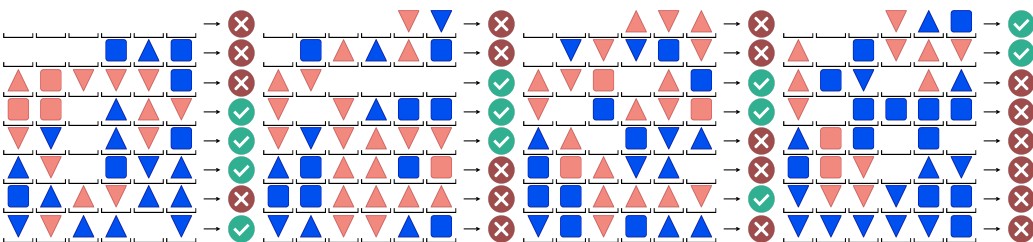

Board 09
Golden rule: "exactly 1 pyramid pointing_up touching red pyramid pointing_down"
**CRN**: "exactly 1 red pyramid pointing_down touching pyramid pointing_up", T-acc 0.95 ✗
**EMP-C**: "exactly 1 red at_the_right_of and red", T-acc: 0.80 ✗
**EMP-R**: T-acc 0.79 ✗

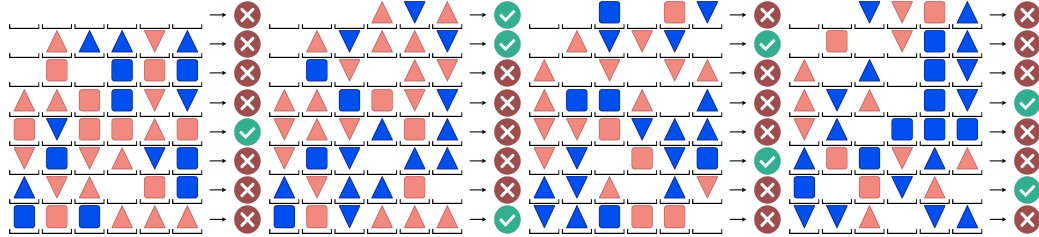

Board 25
Golden rule: "at_least 2 red touching blue pyramid pointing_down"
**CRN**: "at_least 2 red touching blue pyramid pointing_down", T-acc 1.0 ✓
**EMP-C**: "at_least 2 red touching blue pyramid", T-acc: 0.87 ✗
**EMP-R**: T-acc 0.69 ✗

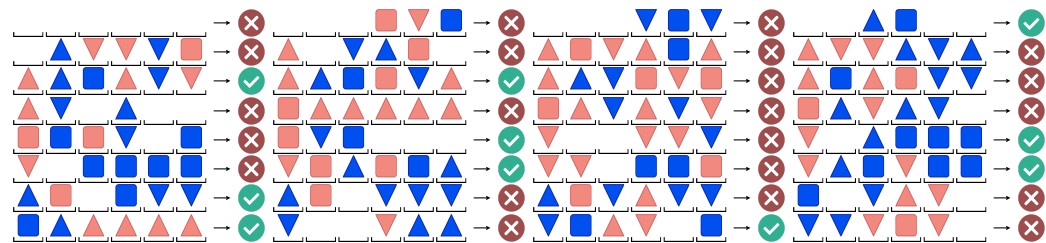

Board 30
Golden rule: "exactly 1 blue pyramid touching blue block"
**CRN**: "exactly 1 blue pyramid touching blue block", T-acc 1.0 ✓
**EMP-C**: "exactly 1 blue pyramid touching block block", T-acc: 0.97 ✓
**EMP-R**: T-acc 0.79 ✗

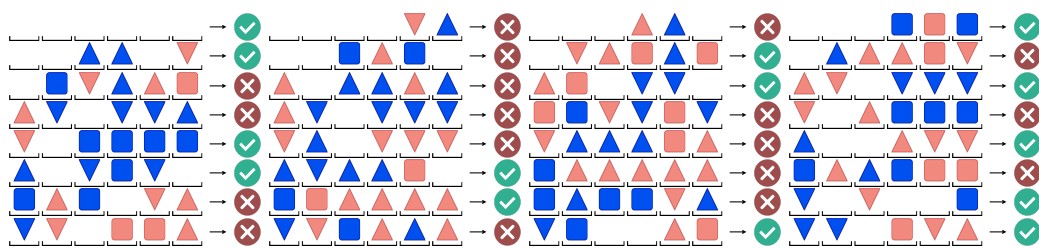

Board 75
Golden rule: "zero blue touching red pyramid"
**CRN**: "zero blue touching red pyramid", T-acc 1.0 ✓
**EMP-C**: "zero blue touching red", T-acc: 0.85 ✗
**EMP-R**: T-acc 0.91 ✓

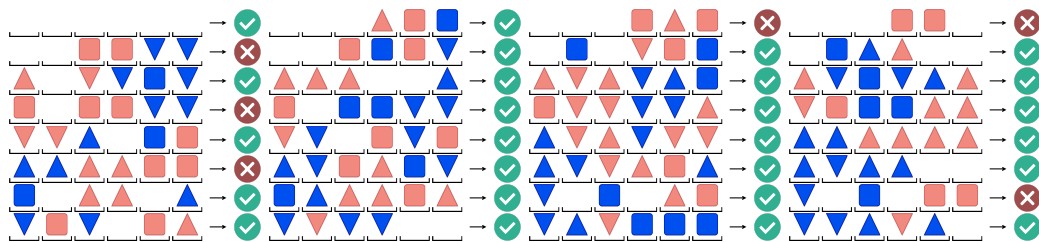

Board 97
Golden rule: "at_most 1 red block touching red"
**CRN**: "at_most 1 red block touching red", T-acc 1.0 ✓
**EMP-C**: "at_most 1 red touching at_the_right_of red", T-acc: 0.98 ✓
**EMP-R**: T-acc 0.93 ✗

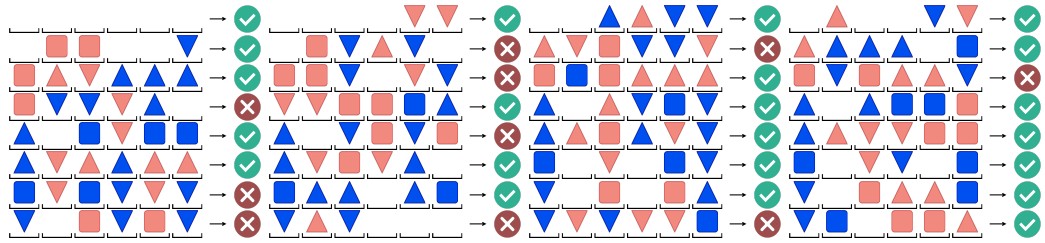

Board 103
Golden rule: "at_most 1 blue pyramid pointing_down touching red"
**CRN**: "at_most 1 blue pyramid pointing_down touching red", T-acc 1.0 ✓
**EMP-C**: "at_most 1 blue pyramid pointing_down touching red", T-acc: 0.98 ✓
**EMP-R**: T-acc 0.85 ✗

