# OpenReview forum: "Explanatory Learning: Beyond Empiricism in Neural Networks"
_ICLR.cc/2022/Conference — ICLR 2022 Submitted_

### Official Review · Reviewer_PMcp · 2021-10-31

**Correctness:** 3
**Technical Novelty And Significance:** 2
**Empirical Novelty And Significance:** 3
**Recommendation:** 6
**Confidence:** 4

**Main Review:**

Strengths:

- Acknowledges the bilateral nature of explanations in building explainable AI/ML.
- Offers a new benchmark problem.

Weaknesses:

- Does not fully explicate the assumptions that it is making in terms of the explanations.
- Does not properly position itself with respect to relevant work in the literature.
- The somewhat over-the-top philosophical discussion distracts from the essential point.

General remarks:

The problem acknowledges the need to have bilateral explainability; i.e., build machines that can explain, by first explaining *to* the machines. This is discussed, for example, by Michael, "Machine Coaching", IJCAI Workshop on XAI 2019, which seems to be rather relevant to the current paper, especially given that the paper makes an effort to connect to learning theory. Another line of work that seems to be relevant is Mozina et al., “Argument Based Machine Learning”, AIJ 2007, where data points are accompanied by an argument (i.e., an explanation) on why they are labeled as they are, as is the case in the current paper. Ignoring the obvious difference from the current work that these two works assume that explanations are in logic, the underlying theme of the cited papers and the current paper seems considerably close to ignore.

I found the philosophical positioning of the paper as a study in empiricism vs rationalism to be somewhat far-fetched. I believe it is instructive to take a step back and seek what are the underlying assumptions of the paper, and whether those bring something new to the table.

It is clear that the label of a data point is a function of the data point and the phenomenon, and not only of the data point. And since the phenomenon is essentially associated with an explanation, then the label is a function of the data point and the explanation. What the authors propose, then, as the pipeline for predicting the label is a form of chaining of learned pieces of knowledge, where one predicts the explanation, and then using that previous prediction one proceeds to predict the label at a second cycle of inferencing. Of relevance here is the work by Valiant, "Robust Logics", AIJ 2000, and Michael, "Simultaneous Learning and Prediction", KR 2014, as well as some follow-up papers by the authors, that establish the benefits of chaining.

From a formal point of view, there is nothing in the explanations that makes them explanations. As the paper says, they are simply strings. Is it really the case that they are arbitrary strings, or should they be strings that are learnable as a function of the data points (or more properly, sets of data points)? That is to say, if each type of explanation for a phenomenon P_j was simply replaced by the number j, would then this still be considered an arbitrary language of explanations, despite having no structure and not being learnable? If indeed, explanations cannot be arbitrary strings, then one needs to carefully state what are the underlying assumptions on the language of explanations. If, on the other hand, explanations can be arbitrary (e.g., explanation j for phenomenon P_j) then this makes the term "explanation" rather mood; it is simply another signal in the data (at a meta-level, see my comment below).

Another aspect of the paper is the two levels of learning problems that exist, as mentioned in the last two paragraphs: the object-level learning problem of learning from a data point and its explanation the label of the data point; and the meta-level learning problem of learning from sets of data points to predict the phenomenon/explanation. Both of these two problems seem to follow the empiricist view, in the sense of the paper, which, as I said above, makes it unclear what the philosophical discussion on empiricism vs rationalism really add to the picture. The conjecture generator seems to be simply a meta-level classifier (albeit a stochastic one). Which brings me back again to the question on whether the explanations need to have some learnable structure.

Additional points:

The metric of NRS seems to include in its definition the identification of a nearest neighbor. Why? Shouldn't the goal be to predict the actual explanation? If the algorithm that makes the prediction wishes to use the nearest neighbor to reach that decision, then that would be fine. But the use of the nearest neighbor seems more natural to be part of the algorithm that attempts to solve the problem, not part of the evaluation metric for measuring success.

The proposed approach to solving the problem seems not to be accompanied by any formal guarantees on its performance. This would typically be compensated by an extensive experimental section, which is not the case here.

I would have found a different narrative for this paper to be more convincing and impactful: the introduction of Odeen as a benchmark problem, and a deeper discussion of its features and parameters, and then present the particular approach, properly placed in the context of relevant work, as a suggested direction for what type of systems would presumably be useful in tackling the Odeen benchmark.

------- After the Author Rebuttal -------

I acknowledge that the authors have made an effort to engage with the points that I raised, and I have increased my score. I believe that the connections with learning theory are much more deep than the brief remarks offered in the revised version of the paper, and I hope that the authors will consider exploring them further in their future work as a formal underpinning of their empirical work.


**Summary Of The Paper:**

The paper considers the problem of classification learning where each data point is accompanied by some explanation. The explanation is in some arbitrary language. The paper proposes a way to increase the performance of the classification task by learning to predict the explanation associated with a given data point, and then using that explanation along with the data point to proceed to make a prediction about the data point's label. The paper implements this pipeline using neural networks, and presents empirical results on a new benchmark dataset to demonstrate its performance.


**Summary Of The Review:**

I find Odeen to be a useful contribution, and one that would raise awareness on the need of certain underused techniques in the machine learning literature. But, the rest of the paper needs to be more clearly and properly placed in the context of existing work in the literature.

---

> ### Author Response · Authors · 2021-11-22
> **We thank reviewer PNcp for the precious references that give us the opportunity to greatly enrich our paper. (1/5)**
>
> We thank reviewer PNcp for the time and effort spent in reviewing our work. We apologize for the late response, which is in part due to the need for an accurate reading of the relevant papers they pointed us to. These references give us the opportunity to greatly enrich our paper.
>
> >The problem acknowledges the need to have bilateral explainability; i.e., build machines that can explain, by first explaining to the machines. This is discussed, for example, by Michael, "Machine Coaching", IJCAI Workshop on XAI 2019, which seems to be rather relevant to the current paper, especially given that the paper makes an effort to connect to learning theory.
>
> We are glad the reviewer caught our point on the bilateral nature of explanations. Training an interpreter using explanations paired with observations (and then using it at test time as described in the algorithm in Figure 3) can be seen as a form of explaining to the machine, which is indeed quite close to the form of interaction described by [1], where “humans explicate their reasoning process, and explain to the machine how some particular tag was derived, or why a certain alternative tag (perhaps one proposed by the machine) is inappropriate.” As stated by the reviewer, there are obvious differences with our work, which reside in the nature of the explanations and their practical execution (natural language vs logic and a learned interpreter vs a human or human-crafted one). Yet, we have added a reference to this relevant work, which also includes a variant of the typical PAC definition [2] that accommodates this bilateral communication which likely could turn useful in future works in this direction.
>
> [1] Loizos Michael, Machine Coaching (2019)
>
> [2] Leslie G. Valiant, A Theory of the Learnable (1984)
>
> >Another line of work that seems to be relevant is Mozina et al., “Argument Based Machine Learning”, AIJ 2007, where data points are accompanied by an argument (i.e., an explanation) on why they are labeled as they are, as is the case in the current paper. Ignoring the obvious difference from the current work that these two works assume that explanations are in logic, the underlying theme of the cited papers and the current paper seems considerably close to ignore.
>
> We have added a reference to the work by Mozina et al. [3] which -apart from the already stated distinctions we discussed for [1]- can be seen as a forerunner of our work: the fundamental observation-explanation pair $(D_i, e_i)$ in our paper resembles the input of the argument-based rule learning algorithm introduced there.
>
> [3] Martin Mozina et al., Argument Based Machine Learning (2007)

---

> > ### Author Response · Authors · 2021-11-22
> > **We thank reviewer PNcp for the precious references that give us the opportunity to greatly enrich our paper. (2/5)**
> >
> > >I found the philosophical positioning of the paper as a study in empiricism vs rationalism to be somewhat far-fetched. I believe it is instructive to take a step back and seek what are the underlying assumptions of the paper, and whether those bring something new to the table.
> >
> > We agree with the reviewer that this kind of work should thoroughly investigate and discuss the assumptions of the underlying idea. This is why we have introduced the Explanatory Learning framework, formalizing the idea of explainability of a phenomenon and of representativity of a set of observations. These are the two core definitions embedded in the EL discussion in section 2 that constitutes the foundations of our work:
> > - **Explainability of a phenomenon**:
> > >a phenomenon $P_i$ is explainable in a language $L$ if there exists a string $e \in \Sigma_{L}$ such that, for any $x \in U$, it occurs $I_{L}(x, e) = 1_{P_i}(x) $, where $1_{P_i}(x)$ is the indicator function over the set $P_i$. We call the string $e$ an explanation, in the language $L$, for the phenomenon $P_i$.
> > The core takeaway here is that phenomena are not explainable per se, but are only explainable with respect to a language. That is, the explainability of a phenomenon depends on the *interpreter* we are considering.
> > - **Representativity of a set of observation**
> > > A dataset $D_0$ is representative for a phenomenon $P_0$ in $L$ if, for every phenomenon $P \neq P_0$ explainable in $L$ there exists at least a $x_i\in D_0$ such that $1_{P_0}(x_i) \neq1_{P}(x_i)$.
> > This definition states when it is possible to say something general about a phenomenon given a set of observations. Again, this possibility depends on the language used. A more expressive language can describe more phenomena but needs more samples to disambiguate between them during the inductive leap, while a smaller language can account for less phenomena but needs smaller datasets to generalize from observations. In the words of Mitchell [4], $L$ is our language of generalizations expressing our necessary biases.
> >
> >
> > We have now put more emphasis on these assumptions, which are already stated in the paper but may have been a bit hidden between the lines.
> >
> > Concerning the philosophical positioning of the paper, we believe that this key to the reading is extremely relevant for our work. Once acknowledged that the objective of a ML task is creating knowledge, we believe that even a small epistemological discussion can enrich the vision of the reader. We believe that ML can take inspiration from epistemology as it is already taking from the neuroscience and psychology literature. Indeed, we are not alone in this thought, since two Truing prizes in AI have recently discussed the epistemological foundations of deep learning, questioning the widespread radical empiricism [5]. Returning to our work, this positioning serves also to point readers to a reference to where we brought these ideas: the test time algorithm of CRNs closely follows the rationalist inference process described by Popper in [6].
> >
> >
> >
> > [4] Tom Mitchell, The Need for Biases in Learning Generalizations (1980)
> >
> > [5] Judea Pearl, Radical empiricism and machine learning research (2021)
> >
> > [6] Karl Popper, The Logic of Scientific Discovery (1935)

---

> > > ### Author Response · Authors · 2021-11-22
> > > **We thank reviewer PNcp for the precious references that give us the opportunity to greatly enrich our paper. (3/5)**
> > >
> > > >It is clear that the label of a data point is a function of the data point and the phenomenon, and not only of the data point. And since the phenomenon is essentially associated with an explanation, then the label is a function of the data point and the explanation. What the authors propose, then, as the pipeline for predicting the label is a form of chaining of learned pieces of knowledge, where one predicts the explanation, and then using that previous prediction one proceeds to predict the label at a second cycle of inferencing. Of relevance here is the work by Valiant, "Robust Logics", AIJ 2000, and Michael, "Simultaneous Learning and Prediction", KR 2014, as well as some follow-up papers by the authors, that establish the benefits of chaining.
> > >
> > > We thank the reviewer for the relevant references. The paper by Valiant [7] represents a pre-neural approach to the problem of learning a general rule from examples and counter-examples. There, “explanations” are logic formulas and inevitably a fundamental assumption is the availability of preprogrammed recognizers for various relationships to parse the examples. We focused on removing this assumption by directly learning these recognizers through pairs $(D_i, e_i)$ using neural networks, following the intuition of Santoro et al. [8] on the relevance of this problem, i.e. building a machine that “interprets something as symbolic on its own rather than simply manipulate things that are only symbols to human onlookers”. Yet, we are glad for this suggestion that allows us to enrich our effort in showing connections to learning theory.
> > > Looking at the other work [9] pointed out by the reviewer and at the joint work of Michael and Valiant [10], we see some connections between the concept of chaining expressed there and our dual inference scheme. The reasoning process underlying these two concepts is not trivial since we are dealing with commonsense knowledge that should be put in a reliable enough form. Indeed, succeeding in this and unlocking the power of reasoning grants a performance improvement in ours and the above-mentioned works. These works enriched our vision of the problem and we hope that the same will occur to the readers, we included both [9, 10] in our related works section.
> > >
> > > [7] Leslie G. Valiant, Robust Logics (2000)
> > >
> > > [8] Santoro et al., Symbolic Behaviour in AI (2021)
> > >
> > > [9] Loizos Michael, Simultaneous Learning and Prediction (2014)
> > >
> > > [10] Loizos Michael and Leslie G. Valiant, A First Experimental Demonstration of Massive Knowledge Infusion (2008)
> > >
> > > >From a formal point of view, there is nothing in the explanations that makes them explanations. As the paper says, they are simply strings. Is it really the case that they are arbitrary strings, or should they be strings that are learnable as a function of the data points (or more properly, sets of data points)? That is to say, if each type of explanation for a phenomenon P_j was simply replaced by the number j, would then this still be considered an arbitrary language of explanations, despite having no structure and not being learnable?  If indeed, explanations cannot be arbitrary strings, then one needs to carefully state what are the underlying assumptions on the language of explanations. If, on the other hand, explanations can be arbitrary (e.g., explanation j for phenomenon P_j) then this makes the term "explanation" rather mood; it is simply another signal in the data (at a meta-level, see my comment below).
> > >
> > > We thank the reviewer for raising this point, which is indeed not immediately clear from the first version of the paper. There is an implicit assumption that we have made explicit in the last version of the paper. Namely that in EL $|\Sigma_L| \gg |A|$ where $|A|$ is the cardinality of the alphabet so that different explanations should necessarily share symbols (and it is not possible to have an explanation $j$ for phenomenon $P_j$). This sharing forces a structure that intuitively should make explanations learnable. Yet, a thorough discussion of the learnability of explanations in EL goes out of the scope of this first paper and will be addressed in a future work about the theoretical foundations of EL.

---

> > > > ### Author Response · Authors · 2021-11-22
> > > > **We thank reviewer PNcp for the precious references that give us the opportunity to greatly enrich our paper. (4/5)**
> > > >
> > > > >Another aspect of the paper is the two levels of learning problems that exist, as mentioned in the last two paragraphs: the object-level learning problem of learning from a data point and its explanation the label of the data point; and the meta-level learning problem of learning from sets of data points to predict the phenomenon/explanation. Both of these two problems seem to follow the empiricist view, in the sense of the paper, which, as I said above, makes it unclear what the philosophical discussion on empiricism vs rationalism really add to the picture.  The conjecture generator seems to be simply a meta-level classifier (albeit a stochastic one). Which brings me back again to the question on whether the explanations need to have some learnable structure.
> > > >
> > > > We thank the reviewer for giving us the opportunity to further clarify this point. For a phenomenon we have data (a set of observations) and theory (the explanation that serves to tag new samples). The end-to-end models (Emp-C and Emp-R) take as input the data and a new sample to output its tag, i.e. they are assuming that the theory needed to tag the sample is already written in data and it just needs to be decrypted, here data comes first (empiricism). On the contrary, in the CRN the tagging of the new sample is performed by an interpreter that takes as input a theory that is already there, and data is used a second time to falsify concurrent conjectures, here theory comes first (rationalism).
> > > > *What about the Conjecture Generator (CG) of the CRN? There the pipeline is empiricist, from data to theory.*
> > > > This is true, but this step is not inescapable in the inference process, conjectures may be chosen at random, or may be already there (communication problem). In CLIP [11] -which can be recognized as an EL approach as discussed in the answer to reviewer ZgwB- this is evident, there is no CG and the interpreter analyzes every possible conjecture. The interpreter is the vital component of the inference process in the CRN.
> > > > Another way of looking at the fundamental difference between the rationalist and empiricist models presented in the paper is given by the nature of the hypothesis space: in CRNs the hypothesis space is the set of all strings $\Sigma_L$, while in the empiricist models the hypothesis space is the n-dimensional vector space $\Theta$, with n equal to the number of parameters of the neural networks implementing Emp-C or Emp-L.
> > > >
> > > > [11] Radford et al., Learning transferable visual models from natural language supervision. (2021)
> > > >
> > > > >The metric of NRS seems to include in its definition the identification of a nearest neighbor. Why? Shouldn't the goal be to predict the actual explanation? If the algorithm that makes the prediction wishes to use the nearest neighbor to reach that decision, then that would be fine. But the use of the nearest neighbor seems more natural to be part of the algorithm that attempts to solve the problem, not part of the evaluation metric for measuring success.
> > > >
> > > > The goal is to predict the actual explanation, and the model’s presented output explanations without needing a nearest neighbor search. This is important since a nearest neighbor search would require to assume to know a priori every possible explanation. As discussed in the paper in paragraph  *Why not explicitly ask for the rule*, our problem is how to evaluate the goodness of the predicted explanations; we can not simply check the output word by word, since we are interested in the meaning to be correct rather than the form. This is why we adopted the zero-knowledge strategy. Yet, we noticed that a simple tag accuracy could be misleading given the goal of predicting the actual explanation for the following reason, also pointed out in section *Metrics*: Given two different rules A and B sharing 99% of the taggings, with A being the correct one, if an EL model tags all the structures according to the wrong rule B, it still reaches an accuracy of 99%, while we want to count this prediction as a full error (the predicted explanation is the wrong one). To solve this problem we designed the NRS metric, which offers a better indication on the correctness of the meaning of the predicted explanations and does so using the privileged information needed to perform NN but unavailable to the models.
> > > >
> > > > [Discussion continues, click on "View 1 more reply"]

---

> > > > > ### Author Response · Authors · 2021-11-22
> > > > > **We thank reviewer PNcp for the precious references that give us the opportunity to greatly enrich our paper. (5/5)**
> > > > >
> > > > > >The proposed approach to solving the problem seems not to be accompanied by any formal guarantees on its performance.
> > > > >
> > > > > There are prominent areas in ML such as PAC learning where it is crucial to provide formal guarantees. At least at this stage, we opted for a more conceptual work focused on the fundamental questions that justify our approach, and chose to position our paper along another line of research, seeing EL more as an improvement over program synthesis, see for instance [12, 13]. Admittedly -as common in this subarea- we did not tackle the problem of discussing formal guarantees in this paper. Yet, we thank the reviewer for having raised this point, since we consider it absolutely essential and we see it as a second natural step in the development of our EL theory, the brief touch to learning theory in the related work section throws us right in this direction.
> > > > >
> > > > > [12] Matej Balog et al. “Deepcoder: Learning to write programs.” (2016)
> > > > > [13] Kevin Ellis et al. "Dreamcoder: Growing generalizable, interpretable knowledge with wake-sleep bayesian program learning” (2020)
> > > > >
> > > > > >This would typically be compensated by an extensive experimental section, which is not the case here.
> > > > >
> > > > > We are open to further suggestions for future works, yet we believe our experimental evaluation to be rather extensive:
> > > > > We evaluated the performance of Emp-C, Emp-R and CRNs on more than one thousand settings of the Odeen environment, comparing their performance using three different metrics (NRS, T-acc and R-acc) on six different compositions of the training datasets (500 or 1,438 rules and 100, 1,000 or 10,000 structures per rule), full results can be seen in Table 3, Appendix B
> > > > > We tested further the generality performance of Emp-C, Emp-R and CRNs on a subset of selected rules with a quantifier (“exactly two”) deliberately excluded from the training set  (*Generalization power* paragraph)
> > > > > We compared the performance of the CRN with a learned interpreter against the CRN with a hard-coded one. Again on more than one thousand settings of the Odeen environment, comparing their performance using two metrics on six different compositions of the training datasets.
> > > > > We tested the Conjecture Generator alone to show how often the correct rule is among the proposed n conjectures on six different compositions of the training dataset, and with n between 1 and 300 (*Adjustable thinking time* paragraph)
> > > > > We compared the performance of the CRN with two different test-time algorithms, the one shown in figure 3 that selects the less contradicted conjecture and the stricter one that selects only conjectures that are never contradicted (*Prediction confidence* paragraph).
> > > > > We showed and measured relevant features of the representations induced by the Odeen language to the Odeen structures (Appendix A, especially Figure 5 and 6).
> > > > > We included a thorough computational cost section that includes a measure of the processing time of all the models on our machines (Appendix C, table 4 and 5).
> > > > >
> > > > > >I would have found a different narrative for this paper to be more convincing and impactful: the introduction of Odeen as a benchmark problem, and a deeper discussion of its features and parameters, and then present the particular approach, properly placed in the context of relevant work, as a suggested direction for what type of systems would presumably be useful in tackling the Odeen benchmark.
> > > > >
> > > > > We are glad the reviewer acknowledges the relevance of the Odeen dataset, which we consider a core contribution of this work and which we hope to release to the AI community contextually with this paper explaining its details and motivation. We really appreciate the suggestion of an alternative editing that wants to put more emphasis on the Odeen environment; we will treasure this suggestion, as it will come useful for future expositions of this work.
> > > > >
> > > > > In hindsight, looking at our first version of the paper now that we have gone through the references reviewer PNcp pointed us to, we realize that we might have given a somewhat incomplete picture of how our contribution positions itself in the current landscape. Nevertheless, we are very satisfied with the last version we arrived at guided by the invaluable suggestions of reviewer PNcp and the others. We really hope reviewer PNcp shares this feeling with us and will consider updating their position.

---

### Official Review · Reviewer_VRSC · 2021-11-02

**Correctness:** 3
**Technical Novelty And Significance:** 2
**Empirical Novelty And Significance:** 2
**Recommendation:** 3
**Confidence:** 4

**Main Review:**

Strengths: 1. The proposed framework forces us to think about explanations and generalization as first class citizens.

Weaknesses:
1. The idea sounds very familiar to explanation based learning. It will be good to contrast the two.
2. My main criticism is that I am not sure we need an entirely new framework for targeting explanation based solutions. Many recent work in NLP try to frame problems in English to get cross task generalization. I would prefer this paper to be positioned as an improvement to such existing approaches, as this is tackling a harder problem class.
3. The experimental results are all on the new game like dataset. It will be good to see performance of proposed methods on real datasets, or already existing synthetic datasets. The baselines are also weak, so its not clear if the proposed techniques will perform better than other cross task generalization methods.

**Summary Of The Paper:**

The paper proposes a new framework for studying explanation driven machine learning problems called Explanatory Learning. The goal is to learn an interpreter model from explanations paired with observations for a particular phenomenon. The explanations might be in an unknown language, but the explanations paired with observations can be used to learn a good interpreter. Once learnt, the interpreter should be able to follow new explanation for an unseen phenomenon. They refer to this problem as the communication problem. They also define the scientist problem where explanations for the unseen phenomenon is not available. The paper also introduces an Odeen dataset to facilitate experiments with Explanatory Learning. The authors propose a neural network architecture as a solution to the scientist problem. It has two components: a conjecture generator which generates a set of candidate English explanations, and a second interpreter model. Experimental results show better generalization compared to end to end neural systems.

**Summary Of The Review:**

Although I am all for explanation based learning solutions, I don't think the components introduced in this paper warrants introduction of a new learning framework. I think it will be better to place it as an improvement to existing approaches for cross task generalization. I also found the experimental results to be weak. It was not clear if the proposed techniques will perform better than other cross task generalization methods.

---

> ### Author Response · Authors · 2021-11-19
> **We believe the raised concerns are readily addressable or are based on misunderstandings. We will try to clarify them in this response. (1/2)**
>
> We thank reviewer VRSC for their feedback and observations. They mainly based their rating on the following criticisms: 1) Missing comparison to "explanation-based learning". 2) EL is similar to recent work in NLP about cross task generalization, we do not need a new framework, and 3) Unclear performance on real datasets and unclear improvement over existing cross task generalization methods.
> As we will detail in the following, we believe these concerns are readily addressable or are based on misunderstandings. We will try to clarify them in this response.
>
> ### 1) Missing comparison to "explanation-based learning".
> Thank you for raising this point and for giving us the opportunity to discuss it. In general, EBL requires the existing knowledge to be expressed through a clean set of axioms describing a formal theory (see for instance [1]), as opposed to the simple strings paired with examples that we instead require in EL. Moreover, EBL suffers from the same problem of ILP, since it needs a human expert that translates the raw data into logic formulas, and the interpreter is not learned. We are in the process of adding the above discussion about EBL vs EL in a concise way in the paragraph “Relationship with other ML problems.”
>
> [1] Minton, Steven. Quantitative results concerning the utility of explanation-based learning.
>
> ### 2) EL is similar to recent work in NLP about cross task generalization, we do not need a new framework.
>
> The problem of cross-task generalization in NLP shares many aspects with the communication and scientist problem of EL, such as the common language in which the different tasks are framed, see for instance [2]. Yet, it exhibits also a crucial difference: in NLP this problem is always in a text-to-text format, i.e. in NLP the problem is framed in a single textual domain where explanations and data share the same role. Conversely, in EL, explanations and data are fundamentally distinct and live in different domains ($\Sigma_L$ and $U$), e.g. text and polygon sequences in this work, text and images in CLIP [3]. We believe that this is a crucial difference, since only with two domains it is possible to define an interpretation operation, which is a map between the textual and the world domains ($\Sigma_L$ and $U$). Also in accordance with the work of Santoro et al. [4, section 3], we emphasized the interpretation operation by designing an entire ML framework around the problem of learning an interpreter.
>
> We do agree with the reviewer that informing the reader about the recent work on cross-task generalization in NLP would enrich the wide picture given by our paper. We will include a brief version of this discussion in the revision.
>
>
> [2] Ye, Qinyuan, Bill Yuchen Lin, and Xiang Ren. CrossFit: A Few-shot Learning Challenge for Cross-task Generalization in NLP. (2021)
> [3] Radford et al., Learning transferable visual models from natural language supervision. (2021)
> [4] Santoro et al., Symbolic Behaviour in AI (2021)

---

> > ### Author Response · Authors · 2021-11-19
> > **We believe the raised concerns are readily addressable or are based on misunderstandings. We will try to clarify them in this response. (2/2)**
> >
> > ### 3) Unclear performance on real datasets and unclear improvement over existing cross task generalization methods.
> >
> > We agree with the reviewer that this work does not offer clear evidence about the performance of CRNs on real-world datasets, as we plan to address this point in future works. Yet, the core contribution of this paper is conceptual in nature, as also acknowledged by reviewer ZgwB. Further, with this work we offer a unified perspective on recent works pushing forward state of the art on real datasets, such as image classification [3], and math word problems [5, 6]. We did so by formalizing the problem of learning an interpreter from observations in the EL framework, by providing an ideal environment to test EL methods (we see Odeen for EL as Atari games for Reinforcement Learning), and by offering a simple rationalist baseline -CRNs- that implements the fundamental tweak we expect to see in order to solve EL problems, but with a rather basic architecture (vanilla transformers trained in the most straightforward way).
> >
> > Concerning the unclear improvement over existing cross task generalization methods -- indeed, we would have liked to show a comparison against existing models, but to the best of our knowledge there exists no method in the NLP literature that can directly tackle the Odeen challenge. In the light of this, we chose to build our own baselines (Emp-C and Emp-R) in the fairest way possible. We adopted the sensible policy of using the same training data and the same architecture with approximately the same number of learnable parameters.
> >
> > We would be very grateful to the reviewer if they can point us to the NLP cross task generalization methods they are thinking of, so that we can try to arrange a comparison against our methods on a proper task in future works.
> >
> >
> > [5] Jianhao Shen, Yichun Yin, Lin Li, Lifeng Shang, Xin Jiang, Ming Zhang, and Qun Liu. Generate & rank: A multi-task framework for math word problems (2021).
> > [6] Karl Cobbe and Vineet Kosaraju and Mohammad Bavarian and Jacob Hilton and Reiichiro Nakano and Christopher Hesse and John Schulman. Training Verifiers to Solve Math Word Problems. (2021)
> >
> > We thank the reviewer again for the constructive feedback. We hope to have dispelled the concerns about our work and to have given them the opportunity to reconsider their position. If not, we are looking forward to embracing any further discussion.

---

### Official Review · Reviewer_H4ra · 2021-11-02

**Correctness:** 3
**Technical Novelty And Significance:** 3
**Empirical Novelty And Significance:** Not applicable
**Recommendation:** 8
**Confidence:** 4

**Main Review:**

Questions:
- In Figure 1, why the first left sequence is correct according to the rule?
- When tagging unseen structures based on the secret rule discovered by the user, how many unseen structures are required to completely rule out other rules that are marginally different from the secret one. Is this number always 1176 in the current setting?
- Since some figures don’t have caption, it’s hard to refer to them… In the figure on the lest side of “Metrics.” How the predicted vector is derived based on various rules?
- As the authors point out the goal here is analogous to that of IPL. Do I understand this correctly that the main advantage of EL to IPL is skipping the translation of the data to logic? If that’s the case, one can learn the translation itself and eliminate the difficulty. Can you elaborate on the advantage of EL over IPL?
- Can’t T-Acc be changes to something auc based instead that takes care of the permissively problem?


Other remarks:
- This is a language specific problem, where order of the sequence does not necessarily matters. The current title does not  reflect this. Perhaps something like Beyond empiricism in neural language learning or something alike would be a better suit.
- Some approaches to explanatory AI that aim at generating explanations as well as predictions such as “TED: Teaching AI to Explain its Decisions” are missed here and should be commended on.

**Summary Of The Paper:**

The paper proposes approaching learning a language as a learning problem that is grammar, alphabet, etc. agnostic. Such formulation has resulted in the so called Explanatory Learning (EL) framework that is paired up with an environment to test it. The starting point for learning a language is to extract an interpreter based on a given set of observation and their explanation and then use the interpreter to determine whether an observation belongs to the language, in a binary classification setting.

**Summary Of The Review:**

- Overall, I quite like the idea of the paper due to the generality of the approach, namely learning an interpreter from observation.

- “proving” the discovered rule is indeed the secret one by examining it in practice (i.e., being used to tag unseen structure) as opposed to revealing the rule directly in text is also a very neat idea.

---

> ### Author Response · Authors · 2021-11-22
> **We thank the reviewer for the positive feedback on the paper. In the following we address their questions and remarks point by point. (1/2)**
>
> We thank the reviewer for the positive feedback on the paper. In the following we address their questions and remarks point by point.
>
> >In Figure 1, why the first left sequence is correct according to the rule?.
>
> The first figure on the left is compatible with the rule, since the fourth moon can be hidden by one of the other moons or by Jupyter itself. The explanation “Four wandering stars having their period around a principal star” is adapted from the English translation of the Sidereus Nuncius [1, page 9 - Examples of Galileo’s drawings starting at page 36]:
>
> >*But what greatly exceeds all admiration, and what especially impelled us to give notice to all astronomers and philosophers, is this, that we have discovered four wandering stars, known or observed by no one before us. These, like Venus and Mercury around the Sun, have their periods around a certain star notable among the number of known ones, and now precede, now follow, him, never digressing from him beyond certain limits. All these things were discovered and observed a few days ago by means of a glass contrived by me after I had been inspired by divine grace.*
>
> We have included in the caption a clarification on why the labeling is correct according to the proposed explanation.
>
> [1] Galilei, Galileo. Sidereus Nuncius. (1610) link to the translation by Albert Van Helden: http://people.reed.edu/~wieting/mathematics537/SideriusNuncius.pdf
>
>
> >When tagging unseen structures based on the secret rule discovered by the user, how many unseen structures are required to completely rule out other rules that are marginally different from the secret one. Is this number always 1176 in the current setting?
>
> We thank the reviewer for pointing this out. The 1176 structures to evaluate the model are chosen so as to uniquely identify the equivalent class of the rule. Actually, to discern between different rules, most of the time, less than thirty labeled structures are needed.
>
> >Since some figures don’t have caption, it’s hard to refer to them… In the figure on the lest side of “Metrics.” How the predicted vector is derived based on various rules?
>
> Given a rule, the predicted vector associated with that rule is calculated evaluating the rule on the structures of the Odeen environment. For each structure the rule produces a labeling (0 or 1), the predicted vector is produced adding on each column (structure) the corresponding label.
>
> >As the authors point out the goal here is analogous to that of IPL. Do I understand this correctly that the main advantage of EL to IPL is skipping the translation of the data to logic? If that’s the case, one can learn the translation itself and eliminate the difficulty. Can you elaborate on the advantage of EL over IPL?
>
> The reviewer here got to the point, the crucial difference between EL and ILP is in the automation of the translation process. Yet, learning this translation is not trivial, and is arguably harder than a natural language translation - which is a difficult problem per se - since an ILP solver does not tolerate any error or ambiguous statement.
> Going deeper, in ILP everything in the world is expressed in logical propositions. These propositions are usually created by an experienced programmer that used his knowledge to encode what and how the observations of the world should be represented (logic formulas). A symbolic ILP solver then uses this knowledge to return the hypothesis in a logic proposition. Conversely, in EL the world is expressed in its natural form (observations),  and a part of the model is responsible for interpreting it in natural language.
>
> The crucial differences of EL with ILP are the following:
> In ILP the logic formula induced by the observed positive and negative predicatives is not learned, the structure of the world is defined by the programmer. Furthermore, both the observations (positive and negative predicatives) and the induced logic formula must be in the same language (logic language).
> In ILP the interpreter is not learned, hence the semantic of the element of the world and of the predicatives is predefined by the programmer.
> As expressed by Santoro et al. [2], we want an “AI that interprets something as symbolic on its own rather than simply manipulating things that are only symbols to human onlookers, and thus will ultimately lead to AI with more human-like symbolic fluency”. This is the problem we want to address with EL.
> [2] Santoro et al. “Symbolic Behaviour in AI”
>
> >Can’t T-Acc be changes to something auc based instead that takes care of the permissively problem?
>
> The main metric used for the Odeen task is Nearest Rule Score (NRS), we added T-Acc as a secondary convenient and familiar score. Results on the dataset (see Appendix D) show that also a high T-acc does not always reflect a correct NRS. In light of this, we opted for a standard accuracy on the fixed 0.5 threshold (T-acc) rather than a more expensive AUC metric taking care also of the permissive problem.

---

> > ### Author Response · Authors · 2021-11-22
> > **We thank the reviewer for the positive feedback on the paper. In the following we address their questions and remarks point by point. (2/2)**
> >
> > >This is a language specific problem, where order of the sequence does not necessarily matters. The current title does not reflect this. Perhaps something like Beyond empiricism in neural language learning or something alike would be a better suit.
> >
> > We think that the proposed method is fairly generic to address a wide range of problems.
> > The centrality of language is only a consequence of the rationalist perspective shift in neural models used to make inferences.
> > In a rationalist setting, we want a hypothesis to be accepted after the falsification of concurrent conjectures using available data. This way, the hypothesis is detached from the model and can only be accepted or refused in toto, in contrast to a model-hypothesis that is continuously updated based on each new data sample.
> > This point constitutes the main motivation of our work and calls for language only in a second moment, when we note how a new conjecture cannot but be expressed in a language, through a novel proposition that is *already* valid in that language; that is why we are able to understand it. This new proposition is an original combination of already known concepts through the rules defining the grammar of our language.
> > These are the main motivations behind the current title.
> >
> > >Some approaches to explanatory AI that aim at generating explanations as well as predictions such as “TED: Teaching AI to Explain its Decisions” are missed here and should be commended on.
> >
> > Thanks for pointing this out. This work makes a good contrast with our approach, we added a reference in the paper. TED [3] recognizes the importance of external human-understandable explanations as opposed to difficult looks at the inner workings of neural models. Yet, differently from CRNs, TED works with language in a chinese-room fashion and does not guarantee a causal link between the explanation and the prediction.
> >
> > [3] Hind et al. “TED: Teaching AI to Explain its Decisions” (2018)

---

### Official Review · Reviewer_ZgwB · 2021-11-05

**Correctness:** 3
**Technical Novelty And Significance:** 2
**Empirical Novelty And Significance:** 2
**Recommendation:** 5
**Confidence:** 2

**Main Review:**

- The paper is a pleasure to read.


- The choice of illustrative examples is excellent.


- The contribution is conceptual in nature, which is good.  Alas, the motivation for pursuing this research direction is unclear.  Specifically:

1) It is not clear why the machine should estimate *both* the link between sentences and data and between sentences and sentences *jointly*.

2) It is not clear why the lanugage would be unknown and unstructured.

3) It is not clear what (new?) real-world applications this setup is meant to capture.

(Please note that this setup does not solve the "semantic gap" problem, because the semantics of the sentences are implicitly supplied by whoever designs the data set.)

I would advise the authors to ground their motivation in concrete conceptual problems and/or applications.

Given that the learning framework is *the* key contribution of the paper (the value of the two other major contributions depends on whether the learning setting makes sense), the fact that motivation is lacking/unclear is a serious concern.

This makes it hard to evaluate the significance of the paper.


- The terminology used in the paper feels somewhat misleading/inappropriate.

As far as I can see, "phenomena" are simply *concepts* and the "explanations" are intensional *descriptions* thereof, except in a language unknown to the machine (but known to the annotator).  I do not understand what is gained by using this terminology.  Indeed, I find the latter confusing and I expect other readers to be confused by it too.

In particular, I do not understand the link between "description" and "explanation", especially considering the causal connotations of the latter term (which are becoming more and more clear as work on causality is being merged into AI and ML.)


- The work seems to rely on a rather strict assumption.  In particular, the fact that D_0 should be discriminative for P_0 in L is (as far as I can see) unlikely to hold in practice -- especially considering that D_0 is supposed to contain a "small set of observations" -- and it simplifies the learning problem considerably.  In what applications is it reasonable for this asumption to hold?  Is it necessary?  What happens if it doesn't hold in practice?  How costly is it to acquire a D_0 that explicitly satisfies this assumption?  Given that such a D_0 would not be IID, how would this impact statistical learning of P_0?  These issues are touched upon in the conclusions, but they deserves an actual discussion.


- The proposed architecture is reasonable but surprisingly involved and the details are hidden in the appendix.  It would be more straightforward to explain in detail the various pieces that make up the architecture (how many, what they take as input and what they spit out) from the get-go, rather than relying on Figure 3, which lacks mathematical precision.


- Inferring whether an instance x satisfies a concept P is surprisingly involved as it requires to generate a (presumably large) number of candidate descriptions for P and then counting, for each description, whether x satisfies it.  Presumably this scales poorly with language complexity.  Is this efficient at all?


- The experiments are limited to an interesting but entirely synthetic (actually, quite toy) setting.  For instance, as far as I can tell no sub-symbolic inputs are present.  Experiments with real data would have been useful to evaluate the efficacy of the proposed pipeline.  CRNs are compared only against two simpler baselines.  I realize that the focus of the paper is in its conceputal contribution, but a more varied selection of experiments would have been welcome.


Minor issues
------------

- Wouldn't it be more natural to define the interpreter as a map from descriptions to classifiers (indicator functions)?

- It would be good to disambiguate the term "explanatory learning" from previous uses of the same term, see "Explanatory interactive machine learning" AIES 2019.

- The "communication problem" shares some aspects with multitask learning, especially so if the language is compositional and the tasks form a hierarchically (or can be related logically to each other, e.g., P0 is the negation of P1 etc.).  It also shares aspects with few-shot learning.  It may be worth highlighting the connection.

**Summary Of The Paper:**

The authors introduce a novel learning framework that revolves around learning a map between concepts and their description, where the latter are expressed in a language unknown to the learner.  Two learning problems are defined: (i) learning a new concept from a description of that concepts as well as examples and descriptions of other concepts, (ii) learn a new concept from examples of that concept as well as examples and descriptions of other concepts.  The underlying assumption is that the map (interpreter) from sentences to sets of examples is shared.  The authors propose an environment to evaluate these tasks and a neural architecture to tackle them.

**Summary Of The Review:**

Potentially great paper with unclear motivation/applications

---

> ### Author Response · Authors · 2021-11-18
> **We are glad the reviewer found the paper enjoyable and potentially impactful, we have tried to address their concerns (1/3)**
>
> We thank reviewer Zgwb for the constructive feedback and the kind comments about the readability and potential of our work. In the following, we discuss their concerns, address all their points, and reference the minor revisions that we are in the process of implementing (the updated paper will be uploaded within a couple of days).
>
> > It is not clear why the machine should estimate both the link between sentences and data, and between sentences and sentences jointly.
>
> The machine Conjecture Generator (CG) estimates the link between observations (data) and explanations (sentences), while the machine Interpreter (I) estimates the correctness of an explanation (sentence) given the observations (data). This double estimate is needed for our test time pipeline. Yet, we are not certain to have answered the question, can the Reviewer kindly elaborate on this point? We are not sure what is meant by “link between sentences and sentences” and what “jointly” refers to. We apologize for the question.
>
> > It is not clear why the language would be unknown and unstructured.
>
> Thank you for sharing this doubt, which we are happy to clarify. In the EL problems we have defined, the language $L$ is unknown to the agent in the sense that the agent is not provided with an interpreter $\mathcal{I}_L$. Therefore, the agent is not immediately able to compare language statements to the real world: these appear to the agent just as mute sequences of symbols, like Japanese strings to a non-Japanese speaker. In contrast, in ILP and Program synthesis we assume to already have an interpreter which turns the abstract logic formulas or programs into actions in the real world. In this sense, in ILP and Program synthesis we assume to work with a language that is already known. Following [1], we believe that this assumption does not hold in many tasks requiring intelligence we want machines to take part in.
>
> We are not sure about what the Reviewer means by “unstructured language”, but we are glad to share our view on structure and language. In our view, a language can only ever be structured (through grammar). Indeed, in EL we do not assume to know explicitly this structure a priori, for instance, we do not assume to have a grammar checker which validates (or not) the form of the conjectures. We just assume to have a set of strings $\{e_1, \dots, e_n \}$ representing explanations of phenomena in a certain language $L$.
>
> >It is not clear what (new?) real-world applications this setup is meant to capture.
> >[...]
> >I would advise the authors to ground their motivation in concrete conceptual problems and/or applications.
> >
> >Given that the learning framework is the key contribution of the paper (the value of the two other major contributions depends on whether the learning setting makes sense), the fact that motivation is lacking/unclear is a serious concern.
> >
> >This makes it hard to evaluate the significance of the paper.
>
>
> Indeed, as the Reviewer acknowledges, our contribution is conceptual in nature. We aim to offer a unified perspective on recent works in AI [1, 2, 3] and work on defining the Explanatory Learning framework around the core idea of learning an interpreter from observations. Yet, these works show the large set of problems and applications that can be tackled by EL: such as image classification [1] or Math Word Problems in NLP [2, 3]. In all these problems the above works are the current SOTA.
> Moreover, we discuss in the paper how EL generalizes Inductive Logic Programming and Program Synthesis. EL removes from ILP and PS the assumption of having a human or human-crafted interpreter that turns output formulas or programs into actual actions in the real world, and from ILP the assumption of having a translator from input data to logic formulas. In this sense, any problem modeled by ILP and PS susceptible to the collection of a dataset is also an EL problem, e.g. automatic text editing or regexes, symbolic regression; see [4 fig. 1 A] for some examples of these and other tasks and applications.
>
> [1] Alec Radford, Jong Wook Kim, Chris Hallacy, Aditya Ramesh, Gabriel Goh, Sandhini Agarwal, Girish Sastry, Amanda Askell, Pamela Mishkin, Jack Clark, et al. Learning transferable visual models from natural language supervision. (2021)
>
> [2] Jianhao Shen, Yichun Yin, Lin Li, Lifeng Shang, Xin Jiang, Ming Zhang, and Qun Liu. Generate & rank: A multi-task framework for math word problems (2021).
>
> [3] Karl Cobbe and Vineet Kosaraju and Mohammad Bavarian and Jacob Hilton and Reiichiro Nakano and Christopher Hesse and John Schulman. Training Verifiers to Solve Math Word Problems. (2021)
>
> [4] Ellis, Kevin, et al. "Dreamcoder: Growing generalizable, interpretable knowledge with wake-sleep bayesian program learning” (2020)

---

> > ### Author Response · Authors · 2021-11-18
> > **We are glad the reviewer found the paper enjoyable and potentially impactful, we have tried to address their concerns (2/3)**
> >
> > >(Please note that this setup does not solve the "semantic gap" problem, because the semantics of the sentences are implicitly supplied by whoever designs the data set.)
> >
> > Thank you for pointing us to the “semantic gap” concept, which is very relevant in our discourse (we find it close to the “barrier of meaning” problem stated by Rota [5, 6]). Without claims of having solved the semantic gap problem, we believe that our work addresses it.
> >
> > Can the reviewer kindly elaborate on the semantics of the sentences implicitly supplied? Assuming that a semantics exists, how can it not be supplied at least implicitly in the dataset? Which different setup would you call as having solved the semantic gap problem?
> >
> > [5] Rota, Gian-Carlo. The barrier of meaning. (1985).
> >
> > [6] Melanie Mitchell, Artificial Intelligence Hits the Barrier of Meaning (2019)
> >
> > >The terminology used in the paper feels somewhat misleading/inappropriate.
> > >As far as I can see, "phenomena" are simply concepts and the "explanations" are intensional descriptions thereof, except in a language unknown to the machine (but known to the annotator). I do not understand what is gained by using this terminology. Indeed, I find the latter confusing and I expect other readers to be confused by it too.
> > >In particular, I do not understand the link between "description" and "explanation", especially considering the causal connotations of the latter term (which are becoming more and more clear as work on causality is being merged into AI and ML.)
> >
> > Yes, we absolutely agree that the terminology “concept” and “description” would make sense. However, we prefer “phenomena” and “explanations” for three reasons:
> > - As the Reviewer has noticed, the term “explanation” carries a causal connotation that we definitely want, but which “description” does not have. The prediction of a CRN is directly caused by the underlying explanation selected (as discussed in the Explainability paragraph in section 5), differently from other approaches like [7], which in our view should admittedly use the term “description” rather than “explanation”.
> > - While it is perfectly fine to slightly adjust “descriptions” to adapt them to the data, this is not the case with explanations (as stressed in the theory of explanations of Deutsch [8], which states that good explanations are hard to vary).
> > - We believe that this choice of terms makes the paper more readable, especially for the philosophy of science and natural sciences community, which we aim to involve in this AI discussion.
> >
> > [7] Hind, Michael, et al. TED: Teaching AI to explain its decisions. (2019)
> >
> > [8] Deutsch David The Beginning of Infinity: Explanations that Transform the World (2011)
> >
> > >The work seems to rely on a rather strict assumption. In particular, the fact that D_0 should be discriminative for P_0 in L is (as far as I can see) unlikely to hold in practice -- especially considering that D_0 is supposed to contain a "small set of observations" -- and it simplifies the learning problem considerably. In what applications is it reasonable for this asumption to hold? Is it necessary? What happens if it doesn't hold in practice? How costly is it to acquire a D_0 that explicitly satisfies this assumption? Given that such a D_0 would not be IID, how would this impact statistical learning of P_0? These issues are touched upon in the conclusions, but they deserves an actual discussion.
> >
> > Thank you for the detailed questions. The assumption mentioned at the beginning of the question is not actually necessary. We made it for simplicity, since in this way the goal is very clear: to predict whether a new sample belongs or not to a single phenomenon uniquely defined by a set of samples $D_0$. Nothing prevents us from providing ambiguous $D_0$ consistent with several phenomena; in that case, we would accept a prediction consistent with any of those phenomena.
> > This assumption holds for instance in many image classification tasks such as ImageNet (successfully solved by CLIP in an EL fashion through an exhaustive search of conjecture [1]), where this assumption guarantees that each picture belong to a single class and only one label is considered true, i.e. we do not have pictures containing both an elephant and a bicycle.

---

> > > ### Author Response · Authors · 2021-11-18
> > > **We are glad the reviewer found the paper enjoyable and potentially impactful, we have tried to address their concerns (3/3)**
> > >
> > > >The proposed architecture is reasonable but surprisingly involved and the details are hidden in the appendix. It would be more straightforward to explain in detail the various pieces that make up the architecture (how many, what they take as input and what they spit out) from the get-go, rather than relying on Figure 3, which lacks mathematical precision.
> > >
> > > The models discussed in the paper ($\mathcal{CG}, \mathcal{I}$, Emp-R, Emp-C) are all built through vanilla encoder-decoder transformers trained as usual. We do not add any intricacy except for the test-time prediction pipeline of the CRN that practically implements the rationalist knowledge acquisition process, which we describe step by step in the pseudocode algorithm in the left of Figure 3.
> > >
> > > To improve the clarity and readability of this part, we will enrich the right part of Figure 3 picturing architectures. We would be grateful to receive any additional suggestions about other architecture details that should be moved from the appendix to the main paper.
> > >
> > > >Inferring whether an instance x satisfies a concept P is surprisingly involved as it requires to generate a (presumably large) number of candidate descriptions for P and then counting, for each description, whether x satisfies it. Presumably this scales poorly with language complexity. Is this efficient at all?
> > >
> > > Please note that this operation is fully parallelizable, thus several conjectures can be tested in the same GPU batch. Tables 3 and 4 in appendix C display the computational cost and the execution time in seconds, which in Odeen remains in the same order of magnitude with respect to the traditional end-to-end approach with t=300 generated conjectures. Moreover, CRNs exhibit a parameter $t$ at test time which controls the trade-off between computational cost and performance. As shown in the inset figure in the “Adjustable thinking time” paragraph in section 5, very cheap models with $t=50$ or even $10$ still express a significant gap in performance vs the traditional empiricist models.
> > >
> > > >The experiments are limited to an interesting but entirely synthetic (actually, quite toy) setting. For instance, as far as I can tell no sub-symbolic inputs are present. Experiments with real data would have been useful to evaluate the efficacy of the proposed pipeline. CRNs are compared only against two simpler baselines. I realize that the focus of the paper is in its conceputal contribution, but a more varied selection of experiments would have been welcome.
> > >
> > > This is true, our contribution is conceptual in nature, future works will be more tailored on applications. Yet, in this paper, we have pointed to existing works on real data that embrace our core idea of learning an interpreter purely from observations, such as CLIP [1] and the ones on Math World problems [2,3].
> > >
> > > >Wouldn't it be more natural to define the interpreter as a map from descriptions to classifiers (indicator functions)?
> > >
> > > We found the idea of an interpreting operation more natural given the relations of our work with program synthesis and the “Barrier of meaning” problem.
> > >
> > > >It would be good to disambiguate the term "explanatory learning" from previous uses of the same term, see "Explanatory interactive machine learning" AIES 2019.
> > >
> > > We inherited the term “explanatory learning” from a 2013 work by Scott Aaronson [9]. Thank you for pointing us to this other interesting work, which differently from EL involves interaction and would be useful to contrast to.
> > >
> > > [9] Scott Aaronson. Why philosophers should care about computational complexity (2013)
> > >
> > > >The "communication problem" shares some aspects with multitask learning, especially so if the language is compositional and the tasks form a hierarchically (or can be related logically to each other, e.g., P0 is the negation of P1 etc.). It also shares aspects with few-shot learning. It may be worth highlighting the connection.
> > >
> > > Thank you for this feedback. We will expand the paragraph “Relationship with other ML problems” to include a brief comparison with Multi-task learning and Few-shot learning.

---

> > ### Comment · Reviewer_ZgwB · 2021-11-29
> > **paper has been improved but some fundamental issues remain**
> >
> > Thank you for updating the paper and apologies for the very late response.  I appreciate the changes you made to the manuscript.
> >
> > Some final observations below.
> >
> > > these appear to the agent just as mute sequences of symbols, like Japanese strings to a non-Japanese speaker.
> >
> > I understand.  My point is that in real-world applications the language that the explanation is encoded in will always be a human language, say English or Japanese.  It is not clear to me what is the advantage of treating it as if it were an alien language when it clearly is not.  I mean, we don't have resources like Wikipedia or pretrained embeddings for Alienese, but we do have them for Japanese.  Why not use them?
> >
> > > Indeed, as the Reviewer acknowledges, our contribution is conceptual in nature. We aim to offer a unified perspective on recent works in AI [1, 2, 3] and work on defining the Explanatory Learning framework around the core idea of learning an interpreter from observations. Yet, these works show the large set of problems and applications that can be tackled by EL: such as image classification [1] or Math Word Problems in NLP [2, 3]. In all these problems the above works are the current SOTA.
> >
> > Unfortunately, I am not sure that I fully appreciate the leap forward.
> >
> > > Can the reviewer kindly elaborate on the semantics of the sentences implicitly supplied? Assuming that a semantics exists, how can it not be supplied at least implicitly in the dataset?
> >
> > Right, I stand corrected.
> >
> > > Yes, we absolutely agree that the terminology “concept” and “description” would make sense. However, we prefer “phenomena” and “explanations” for three reasons:
> >
> > I tend to disagree, for two reasons:
> > - Explanations are often partial and/or approximate, while here the whole point is (approximately) guess a complete, exact description.  The supervision is complete and exact (albeit encoded in an unknown language).
> > - Phenomena are linked to physics (and to subjective perception, for obvious reasons), which needs not be the case here.  "Concept" is a well-established term in AI and refers to all sorts of classes and groupings of elements.
> > Hence I still find the choice of terminology confusing and I estimate others in the ML community to face the same difficulty as I do.
> >
> > > Thank you for the detailed questions. The assumption mentioned at the beginning of the question is not actually necessary. We made it for simplicity, since in this way the goal is very clear
> >
> > The text should be rephrased so not to mention this assumption or to explicitly lift it once it is no longer useful, otherwise it is simply confusing (it definitely confused me, after all).
> >
> > > The models discussed in the paper (, Emp-R, Emp-C) are all built through vanilla encoder-decoder transformers trained as usual. We do not add any intricacy except for the test-time prediction pipeline of the CRN that practically implements the rationalist knowledge acquisition process, which we describe step by step in the pseudocode algorithm in the left of Figure 3.
> >
> > The reason why I pointed this out is that the lack of a proper formal description of the architecture in the main text hurts readability.  As a reader, I am always disappointed when I am forced to look elsewhere for a key piece of information -- it is a waste of my precious (to me) time.  I expect other readers to think the same.  I think that a formal description would benefit the main text.  Still, I appreciate the more detailed figure.
> >
> > > Please note that this operation is fully parallelizable, thus several conjectures can be tested in the same GPU batch. Tables 3 and 4 in appendix C display the computational cost and the execution time in seconds, which in Odeen remains in the same order of magnitude with respect to the traditional end-to-end approach with t=300 generated conjectures. Moreover, CRNs exhibit a parameter  at test time which controls the trade-off between computational cost and performance. As shown in the inset figure in the “Adjustable thinking time” paragraph in section 5, very cheap models with  or even  still express a significant gap in performance vs the traditional empiricist models.
> >
> > This is good to know.  Still, this does not address the fact that inference -- compared to a common feed-forward architecture -- is much more involved.  I expect future developments of this idea to revolve around a more direct procedure.
> >
> > > This is true, our contribution is conceptual in nature, future works will be more tailored on applications. Yet, in this paper, we have pointed to existing works on real data that embrace our core idea of learning an interpreter purely from observations, such as CLIP [1] and the ones on Math World problems [2,3].
> >
> > I believe that a more clear link between CLIP and novel applications would help immensely to motivate the proposed framework.

---

> > > ### Author Response · Authors · 2021-11-30
> > > **We are glad the reviewer appreciated the changes made to the manuscript. We would like to elaborate further about the unknown language, we feel that here there is a misunderstanding and this one is a core point of our work.**
> > >
> > > We are glad the reviewer appreciated the changes made to the manuscript. We would like to elaborate further about the unknown language, we feel that here there is a misunderstanding and this one is a core point of our work.
> > >
> > > The language is not unknown because it is alien, it is unknown because we want to model the problem of mastering a language currently unknown to the agent. We want to model the process of a non-Japanese speaker that learns Japanese, or of a medicine freshman that learns to read and write technical prescriptions.
> > > We believe that this problem cannot be escaped when we want to create a machine capable of formulating new hypotheses for unexplained observations. That is, we believe that injecting our own knowledge of the semantics of a language would lead to machines that generate symbols that are meaningful only to human onlookers, not to the machine itself.
> > >
> > > As we discuss in the "relationship to other ML problems" paragraph, in real-world applications solving the language-mastering problem removes the need of an expert that:
> > > 1. Interprets the messages for the machine. E.g. in Odeen a compiler that translates rules like "at least one pyramid" to regular expressions that match patterns in the strings representing structures. EL do not need it, the interpreter is learned
> > > 2. Interprets the output of the machine. E.g. In Odeen the output explanation, that CRNs directly interprets outputting a tag for the query structure.
> > >
> > >
> > > A more detailed discussion of this point can be found in [1] and in chapter 6 of [2], titled "The location of meaning".
> > >
> > > [1] Santoro et al. "Symbolic behaviour in AI" (2021)
> > >
> > > [2] Hofstadter "Gödel, Escher, Bach: an Eternal Golden Braid" (1979)

---

> > > > ### Comment · Reviewer_ZgwB · 2021-11-30
> > > > **Brief comment**
> > > >
> > > > > We believe that this problem cannot be escaped when we want to create a machine capable of formulating new hypotheses for unexplained observations.
> > > >
> > > > We have fundamentally diverging opinions on this point.  Injecting our own knowledge is what we do with our offspring in order to be able to bi-directionally communicate with them.  I don't see any downside to injecting bias into machines if this makes them more introspectable/interactable.  Similar points are in a recent paper:
> > > >
> > > >   Kambhampati, Subbarao, et al. "Symbols as a lingua franca for bridging human-ai chasm for explainable and advisable AI systems." arXiv preprint arXiv:2109.09904 (2021).
> > > >
> > > > accepted as a blue sky paper at AAAI'22, if I remember correctly.  I tend to agree with the points raised therein.
> > > >
> > > > Moreover, I find it hard to believe that:
> > > >
> > > > > injecting our own knowledge of the semantics of a language would lead to machines that generate symbols that are meaningful only to human onlookers, not to the machine itself
> > > >
> > > > It is hard to prove that a machine is less likely to "understand" a human language than it is to "understand" an invented language -- especially given that we cannot directly access the semantics of the latter.
> > > >
> > > >
> > > > Regardless, this (very interesting) discussion is independent of the quality of the manuscript itself -- which I think is good, except for the flaws that I pointed out already.

---

> > > > > ### Author Response · Authors · 2021-11-30
> > > > > **We wish to generate interest in this discussion among as many ML researchers as possible**
> > > > >
> > > > > We thank the reviewer for the relevant reference and for engaging in the fascinating discussion about the nature of language learning. This is indeed one of the main goals of our paper; we wish to generate interest in this discussion among as many ML researchers as possible.
> > > > > We believe that we addressed the other points raised by the reviewer in the last version of the manuscript.
> > > > > If that is the case and the reviewer thinks that our paper is good, we hope they consider updating their evaluation.

---

### Author Response · Authors · 2021-11-22
**To All Reviewers: Updated Manuscript**

We thank all the Reviewers for their detailed and constructive feedback. We are glad to see that the Reviewers appreciated the potential impact of our contribution, the pleasant writing, and the novel benchmark and dataset that come with our paper. At the same time, we acknowledge that more critical aspects have also been identified, most notably a somewhat unclear positioning and the lack of experiments on real-world data. Nonetheless, we are happy to confirm that we can resolve these issues within a minor revision, and in the following we explain in detail how we are addressing these points.

In the light of this, to directly address the issues raised in the reviews, we have modified the paper as follows:

- **@Reviewer ZgwB**: In the related work section we clarified the differences between our work and “Explanatory interactive machine learning” by Teso and Kersting.
- **@Reviewer H4ra**: In the related work section we clarified the differences between our work and “TED: Teaching AI to explain its decisions” by Hind et al..
- **@Reviewer ZgwB**: We added a brief discussion on EL vs multitask learning and few-shot learning in the “Relationship with other ML problems.” paragraph.
- **@Reviewer ZgwB**: We updated Figure 3, which now shows more details about the architecture discussed in a more extensive caption.
- **@Reviewer VRSC**: We added a brief discussion on EL vs Explanation-based learning in the “Relationship with other ML problems.” paragraph.
- **@Reviewer VRSC**: We added a brief discussion on EL vs Cross-task generalization in NLP in the “Relationship with other ML problems.” paragraph.
- **@Reviewer H4ra**: We included a clarification on why the labeling is correct according to the explanation proposed in the caption of Figure 1.
- **@Reviewer H4ra**: We added a reference to “TED: Teaching AI to Explain its Decisions” by Hind et al. in the paragraph “Explainability” in the Experiments section.
- **@Reviewer H4ra**: We cleared which is the predicted vector in the “Metrics” paragraph.
- **@Reviewer PNcp**: We have expanded our related work section on Leanring theory, including a whole new discussion with references to [1, 2, 3, 4, 5].
- **@Reviewer PNcp**: We explained further an implicit assumption about the structure of explanations in the “Problem setup” section.

We encourage the Reviewers to compare the new version of the paper vs the old one using the powerful "Show revisions" instrument of OpenReview (right down the title).

In the following, we give detailed responses to all the questions of the individual Reviewers.

[1] Leslie G. Valiant, Robust Logics (2000)

[2] Loizos Michael and Leslie G. Valiant, A First Experimental Demonstration of Massive Knowledge Infusion (2008)

[3] Loizos Michael, Simultaneous Learning and Prediction (2014)

[4] Martin Mozina et al., Argument Based Machine Learning (2007)

[5] Loizos Michael, Machine Coaching (2019)

---

### Decision · Program_Chairs · 2022-01-20

**Decision:**

Reject

**Comment:**

This paper presents an explanation-based learning approach that learns from both observations (examples) and explanations paired with examples. It proposes to learn an interpreter that can map from natural language sentences to examples. The authors also develop an evaluation environment and protocols for the tasks.

Strengths:
- The proposed idea is intuitive and seems general
- The benchmark dataset is useful resource for future research

Weakness:
- The motivation of the present problem setup needs more justification
- The close connection to the line of work on explanation-based learning (especially recent ones in modeling natural language explanations) are not thoroughly discussed and compared.
- Experiments beyond the game-like datasets will help validate the claims better and justifies that the problem setup has real-world applications